# Oncogenic mutations of KRAS modulate its turnover by the CUL3/LZTR1 E3 ligase complex

Andreas Damianou[1,2,]*, Zhu Liang[1,2,]*, Frederik Lassen[1,3], Iolanda Vendrell[1,2], George Vere[1], Svenja Hester[1], Philip D Charles[1,2], Adan Pinto-Fernandez[1,2], Alberto Santos[3,4,5], Roman Fischer[1,2], Benedikt M Kessler[1,2]

**KRAS is a proto-oncogene encoding a small GTPase. Mutations contribute to ~30% of human solid tumours, including lung adenocarcinoma, pancreatic, and colorectal carcinomas. Most KRAS activating mutations interfere with GTP hydrolysis, essential for its role as a molecular switch, leading to alterations in their molecular environment and oncogenic signalling. However, the precise signalling cascades these mutations affect are poorly understood. Here, APEX2 proximity labelling was used to profile the molecular environment of WT, G12D, G13D, and Q61H-activating KRAS mutants under starvation and stimulation conditions. Through quantitative proteomics, we demonstrate the presence of known KRAS interactors, including ARAF and LZTR1, which are differentially captured by WT and KRAS mutants. Notably, the KRAS mutations G12D, G13D, and Q61H abrogate their association with LZTR1, thereby affecting turnover. Elucidating the implications of LZTR1-mediated regulation of KRAS protein levels in cancer may offer insights into therapeutic strategies targeting KRAS-driven malignancies.**

## Introduction

Kirsten rat sarcoma virus (KRAS) is a signal-transducing proto-oncogene that belongs to the superfamily of small GTPases (Gimple & Wang, 2019). KRAS, HRAS, NRAS, and other RAS family proteins act as molecular switches that alternate between active and inactive conformational states, determined by their binding to GTP or GDP. Upon environmental stimuli that induce growth factor-mediated signalling, guanine nucleotide-exchange factors (GEFs), such as RASGEF1A and RASGRF2, are recruited to facilitate the exchange of GDP to GTP, leading to KRAS activation (Buday & Downward, 1993; Vigil et al, 2010). Active KRAS promotes signal transduction through effectors such as RAF

(Moodie et al, 1993), TIAM1 (Lambert et al, 2002), and PI3K (Castellano & Downward, 2010), and persists until GTP is hydrolysed to GDP, which is catalysed by KRAS-associated GTPase-activating proteins.

A balanced equilibrium exists between active and inactive KRAS. In mammalian cells, this can be disrupted by activating point mutations in KRAS, leading to neoplastic properties (Prior et al, 2012). These mutations activate chronic KRAS signalling, either by impairing GTPase activity or affecting KRAS's interaction with negative regulators (Liu et al, 2019). Subsequent constitutive activation of pro-neoplastic signalling leads to uncontrolled cell division and malignant transformation (Haigis, 2017), a trait that can be blocked by small molecules that interfere with GTP/GDP exchange mechanisms (Kim et al, 2023).

Despite these detailed molecular insights into KRAS as a GTPase, its turnover and mutation-driven effects on KRAS interactors remain incompletely understood. Proximity labelling techniques, such as BioID, TurboID, and ascorbate peroxidase 2 (APEX2), recently developed to capture a protein's molecular and cellular proximity proteome (Trinkle-Mulcahy, 2019), were applied to WT KRAS. For instance, BioID-based studies profiled the RAS proximity proteome, unravelling mTORC2 as a direct KRAS effector (Kovalski et al, 2019), and LZTR1, a ubiquitin E3 ligase adaptor, regulating KRAS ubiquitination (Bigenzahn et al, 2018). LZTR1, as part of the CUL3 E3 ubiquitin ligase complex, was proposed to function as a "Ras killer protein" through polyubiquitination and degradation of endogenous RAS via the ubiquitin-proteasome pathway (Abe et al, 2020). In addition, LZTR1/CUL3 was demonstrated to trigger the ubiquitination of HRAS at position K170, thereby altering its cellular localisation by attenuating the interaction of HRAS with the membrane (Bigenzahn et al, 2018). Mono-ubiquitylation-dependent membrane association of NRAS and KRAS is also affected by the deubiquitylase OTUB1 (Baietti et al, 2016). RAS mono-ubiquitylation may also affect other functions, such as the restoration of the GEF-mediated nucleotide exchange of KRAS via modification at residue K104 (Yin et al, 2020).

[1]Target Discovery Institute, Centre for Medicines Discovery, Nuffield Department of Medicine, University of Oxford, Oxford, UK [2]Chinese Academy for Medical Sciences Oxford Institute, Nuffield Department of Medicine, University of Oxford, Oxford, UK [3]Big Data Institute, Nuffield Department of Medicine, University of Oxford, Oxford, UK [4]Center for Health Data Science, Faculty of Health Sciences, University of Copenhagen, Copenhagen, Denmark [5]NNF Center for Protein Research, Faculty of Health Sciences, University of Copenhagen, Copenhagen, Denmark

Correspondence: andreas.damianou@ndm.ox.ac.uk; benedikt.kessler@ndm.ox.ac.UK
George Vere's present address is MRC Medical Centre for Medical Mycology, University of Exeter, Exeter, UK
*Andreas Damianou and Zhu Liang contributed equally to this work

Whereas previous studies mainly focused on WT KRAS, our study explores how oncogenic mutations affect the dynamic molecular interactions of KRAS mutants. To this end, we applied an APEX2-based rapid proximity labelling approach to compare the interactomes of WT KRAS with three mutants G12D, G13D, and Q61H upon starvation and/or acute stimulation by exposure to FCS to track the dynamic associations of active and inactive forms. Quantitative mass spectrometry analysis confirmed previously reported KRAS interactors such as RAF and LZTR1. Remarkably, the WT KRAS proximal proteome was altered strongly between the starvation (GDP-bound form) and the FCS-induced (GTP-bound), a less pronounced trait with mutant counterparts. Proximal proteome analysis revealed that LZTR1 was one of the most predominant proteins interacting differentially between WT and KRAS mutants. Biochemical studies uncovered that WT KRAS and LZTR1, the latter as part of the CUL3 ubiquitin E3 ligase complex, affect each other's protein stability, revealing a direct feedback loop mechanism. KRAS activation mutations appear to alter this regulatory circuit, potentially contributing to aberrant signalling in cancer cells.

## Results

### KRAS-APEX2 WT, G12D, G13D, and Q61H mutants elicit similar expression, localization, and signalling properties as endogenous KRAS

To compare the proximal proteome between the WT KRAS and three of the most frequent oncogenic driver mutants (G12D, G13D, and Q61H), we established an APEX2-dependent proximity labelling method (Fig 1A). We first generated tetracycline-inducible stable cell lines to minimise KRAS misfolding and mislocalization. The same parental cell line was used as a control to develop a "benchmark" proteome as a reference. Four different HEK293 FRT T-Rex cell lines, KRAS WT, G12D, G13D, and Q61H, respectively, expressing WT KRAS- and all mutant-APEX2 fusion proteins in a stable fashion upon treatment with tetracycline, were generated (Fig 1B). KRAS-APEX2 fusions were expressed at sixfold to eightfold the level of the endogenous isoform (Fig S1A). To examine the activity of APEX2 within the expressed constructs, SDS–PAGE/streptavidin blot analysis and immunofluorescence were used to test their in-cell biotinylation ability. In the presence of tetracycline, phenol biotin, and $H_2O_2$, biotinylated proteins were detected as a wide-band pattern. In contrast, only three endogenous biotinylated protein bands were observed in negative controls, indicating high specificity of APEX2 labelling (Fig S1B).

APEX2 tagging may interfere with cellular localisation, signal transduction, and GTP hydrolysis. To evaluate this, we examined ERK/phospho-(p)-ERK activation in all four cell lines after starvation for 2 and 4 h, respectively. As expected, pERK levels were reduced upon starvation in WT KRAS-expressing cells. However, this reduction was not observed in the cells expressing mutant variants (Fig 1B). GTP-KRAS levels, measured via GST-RAF1-RBD enrichment, were decreased upon starvation only in cells expressing WT but not in the constitutively active KRAS mutants, indicating that APEX2 does not interfere with these processes (Fig 1B). KRAS-APEX2 fusion

protein expression was further examined for its localisation. To this end, T-Rex tetracycline-inducible cell lines expressing GFP-KRAS fusion protein and GFP only were generated (Fig S1B). Microscopy analysis showed that GFP-tagged KRAS was localised on the plasma membrane, whereas GFP was mainly diffusing in the cytosol (Fig S1C). Residual nuclear localisation was noted in less than 10% of the cells (Fig S1C). Furthermore, subcellular fractionation was used to separate membranes from cytoplasmic fractions, confirming the localisation of KRAS-GFP and GFP only as observed by microscopy (data not shown). Also, the KRAS-APEX2 WT and mutant fusion proteins were enriched in the membrane fraction (Fig S1D). We also examined the ability of KRAS-APEX2 fusion proteins to translocate upon starvation and FCS stimulation, as its endogenous counterpart does. A sucrose gradient fractionation experiment showed similar translocation and localisation patterns for KRAS-APEX2 fusion protein and endogenous KRAS upon 10-min FCS stimulation, followed by 15 h after initial starvation (Fig S1E). Altogether, the above results suggest that KRAS-APEX2 fusion proteins showed similar functional properties as endogenous KRAS in terms of subcellular localisation.

### APEX2 labelling reveals differential molecular environments between WT and mutant KRAS

Proximity labelling was conducted in HEK293 cells expressing APEX2-tagged KRAS in its WT, G12D, G13D, and Q61H forms to compare their molecular vicinities. For control purposes, the APEX2-KRAS WT-expressing cell line was analysed with and without tetracycline induction. We did not use APEX2-only as a control cause its cytosolic localisation may cause bias towards specific targets. Cells were treated with tetracycline overnight for sufficient expression of the bait, followed by incubation in the presence or absence of FCS for 15 h (Fig 1C). Protein biotinylation was observed only in the presence of tetracycline. During starvation, the Raf/MEK/ERK pathway remained inactive in WT cells, in contrast to the cells expressing oncogenic mutants, where the pathway was constitutively active (Fig 1D). Afterwards, biotinylated material was enriched as shown in Fig 1E. It was then normalised to ensure similar quantity before digestion and analysed using quantitative mass spectrometry. Using traditional volcano blot analysis in this experimental approach is impractical because of the high background noise inherent to APEX2 proximity labelling methodology (Fig S2A). As a result, MS data were further processed with an interaction scoring algorithm, SAINTexpress (Teo et al, 2014), to establish "interaction" probabilities (true proximity proteins) with KRAS baits under different conditions. A SAINT score threshold of 0.8 with a ≥ fourfold enrichment yielded a total of 1,373 KRAS proximal proteins with 700–1,000 proteins per condition (Fig 2A, Table S1).

First, KRAS vicinity partners were divided into several categories, including receptors, activators, repressors, interactors, and canonical RAS effector proteins (Kiel et al, 2021). In addition, to "benchmark" our results, the BioGRID (https://thebiogrid.org/) database was used to retrieve the previously known KRAS "proximitome" (Adhikari & Counter, 2018; Bigenzahn et al, 2018). We detected proteins associated with KRAS signal transduction (receptor tyrosine kinases, including EGFR, MET, and ERBB2IP), eight

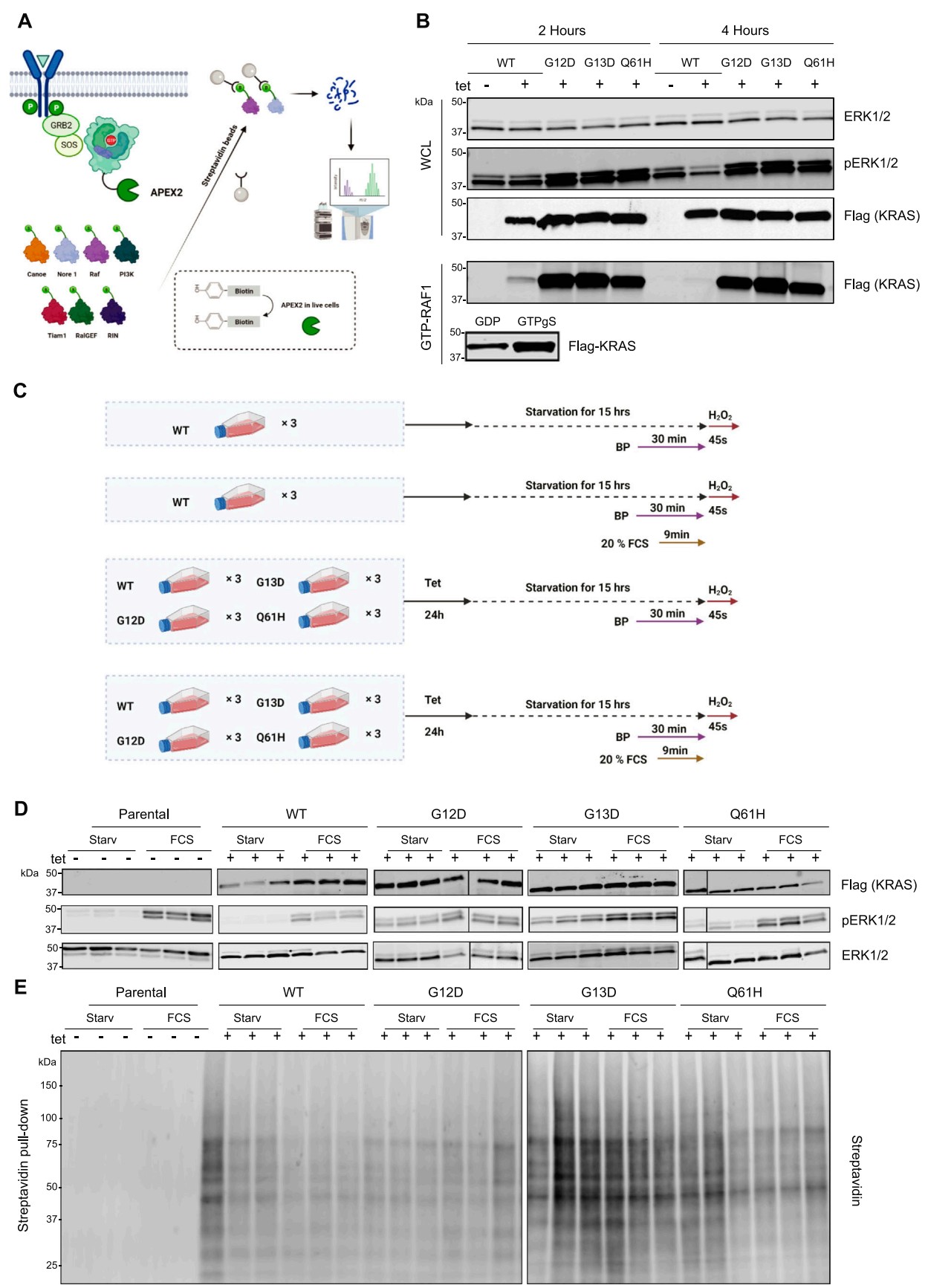

Ras effectors (RAF1, ARAF, TIAM1, RAPH1, RAPGEF2, RAPGEF2, SNX27, and MYO10) (Fig 2B and C). Also, several known KRAS interactors, activators, and repressors were identified in our experiment, including the E3 ligase adaptor proteins LZTR1, DAG1, and GRB2 (Fig 2B and C). We also captured several proteins found to be part of downstream KRAS signalling, including proteins involved in the networks of MEK-Raf (3 proteins), PIK3 (8), mTOR (2), TIAM-RAC (3), RALGDS (8), and RASSF pathways (1) (Fig 2B, Table S1).

206 of the 1,373 proximal proteins identified in our study were also found in previous BioID proximity labelling studies (Fig 2B). Here, we aimed to systematically search for partners associated with KRAS that have different proximal abundance profiles when interacting with various KRAS mutants. We selected both canonical and non-canonical interactors, including effectors, activators, repressors, and receptors, and visualised them (Fig 2C). Of the 105 previously reported physical interactors, 22 (21%) were identified in this study. Notably, LZTR1, ARAF, RAF1, and NF1 showed different abundance profiles between WT and mutant KRAS (Fig 2C). APEX2 proximity labelling efficiency is reflected by the capture of low-abundance proteins that are not readily present in HEK293 deep proteome data sets (Geiger et al, 2012). For instance, we captured several low-abundance canonical KRAS "interactors," such as RAF, TIAM1, SNX27, and MYO10, demonstrating effective APEX2-based enrichment (Figs 2D and S2C).

Despite stringent threshold criteria, 1,373 proteins remained after the first pass, which requires further filtering approaches for the confident discovery of novel KRAS "proximitome" candidates. We then decided to compare the "proximitome" of the different KRAS mutants with the WT, focusing on the specific alterations in the microenvironment associated with each KRAS mutant. This potentially eliminates confounding factors originating from shared backgrounds. Different from previous studies, our work highlights how the oncogenic mutations G12D, G13D, and Q61H, located in the KRAS G domain, perturbed the KRAS molecular environment. For instance, in contrast to WT KRAS, RAF1, and ARAF were enriched with G12D, G13D, and Q61H in both starvation and FCS stimulation (Figs S2D and S3A–F), suggesting that the RAF-MEK-ERK pathway is highly activated in these mutants, consistent with the constitutively phosphorylated ERK signalling (Fig 1D) and previous studies (Haigis, 2017). It is noted that not only the mutations but also the expression level can affect protein-protein interaction networks. Therefore, the enhanced interaction between ARAF, RAF1, and KRAS mutants may also be caused by high expression. In addition, NF1 and SPRED2 (starvation) are only enriched with the G13D and Q61H mutants but

not with G12D, suggesting that G13D and Q61H are associated with NF1/SPRED2 (Figs S2E and S3A–E) to a greater extent as compared with G12D, consistent with previous findings reporting that NF1 is not able to affect G12D KRAS (Rabara et al, 2019). Notably, we found that LZTR1, an essential regulator for KRAS ubiquitination and degradation, was less abundant in all three mutant KRAS proximal proteomes, suggesting a decreased interacting affinity with mutant KRAS (Figs S2E and S3A–E).

To further investigate the aberrant signalling pathways associated with different KRAS mutants, a network enrichment analysis was performed using Metascape (Zhou et al, 2019). The network showed that several processes were shared between resting and FCS stimulation conditions (Fig 3A). However, under resting conditions, specific pathway clusters were mainly shared between the G13D and Q61H mutants, indicating that mutations affect KRAS function through perturbations in proximal protein networks (Fig 3A). In contrast, under FCS stimulation conditions, most pathway clusters were shared among all three mutants, suggesting less mutant-specific effects (Fig 3B).

Finally, we examined hits that were previously identified in other proximity studies as well as in our experiment, focusing on how they changed between the WT and the three KRAS mutants. As noted, we identified proteins that showed specific affinity towards KRAS mutants. For example, mTOR shows a stronger association with all three mutants as compared with WT KRAS, and conversely, NF2 has a stronger association with WT compared with mutants (Fig 3C). We examined whether unique proximal proteins identified exclusively in our study, such as CALR, EIF1AX, and GOLGB1, exhibit variations in the proximity of the oncogenic KRAS mutation (Fig 3D).

## LZTR1 differently affects WT versus KRAS mutants

LZTR1 demonstrated weaker association with oncogenic G12D, G13D, and Q61H KRAS mutants compared with WT (Figs 4A and S3A–E). Furthermore, we explored whether alterations in the proximal microenvironment of KRAS physical interactors stemmed from protein level up-regulation or down-regulation. We conducted proteomic analysis on our cell lines (WT, G12D, G13D, and Q61H) under starvation conditions and monitored protein abundances to ascertain that identified proximal alterations were not contingent on protein level variations (Table S2). When comparing the proteome of the three oncogenic mutants to the WT, several proteins exhibited alterations, including KRAS itself (Figs 4B and S3F and G, Table S2). Interestingly, most of the known KRAS interactors found

**Figure 1. KRAS-APEX2 WT, G12D, G13D, and Q61H mutants express, localise, and function similarly to endogenous KRAS.**
**(A)** Schematic representation of the approach used in this study to investigate the proximal proteome of K-Ras. The first step is the generation of a fusion K-Ras APEX2 protein with a flexible linker between K-Ras and APEX2. In the second step, phenol biotin and $H_2O_2$ were added to the media to facilitate the biotinylation of proteins proximal to the bait, including effectors, regulators, and interactors of K-Ras and proteins located near K-Ras. The next step included the IP using streptavidin beads to capture biotinylated proteins. Finally, biotinylated proteins were digested, and mass spectrometry was used to identify proximal proteins. **(B)** Western blot analysis showing ERK, pERK, and a-FLAG in both cell extract and IP (immunoprecipitation) using GST-Raf1-RBD (Activate Ras Detection Kit) 2 and 4 h upon starvation. As an IP-positive control, the lysate was incubated with GTPγS, and for an IP-negative control, the lysate was incubated with GDP. **(C)** Experiments were carried out in biological triplicates (n = 3). KRAS has a long half-life; the absence of tetracycline will decrease the proximal labelling of the newly expressed KRAS. For the starvation condition, cells were directly moved for 30 min of incubation with phenol biotin and subsequent incubation with $H_2O_2$ for 45 s. On the other hand, for the FCS induction condition, cells were starved for 15 h with a subsequent incubation with phenol biotin for 30 min. In the last 10 min, an additional 20% FCS (including biotin phenol) was added to activate the cells. 45 s of $H_2O_2$ was used to initiate the biotinylation of the KRAS proximal proteins. **(D)** Western blot analysis showing ERK, pERK, and a-FLAG in-cell extract upon APEX2 labelling under different environmental conditions, including starvation and FCS induction. **(E)** Western blot analysis indicated biotinylated proteins (streptavidin, DyLight 488 conjugated) upon streptavidin immunoprecipitation. APEX2 experiments were carried out in three independent biological experiments.

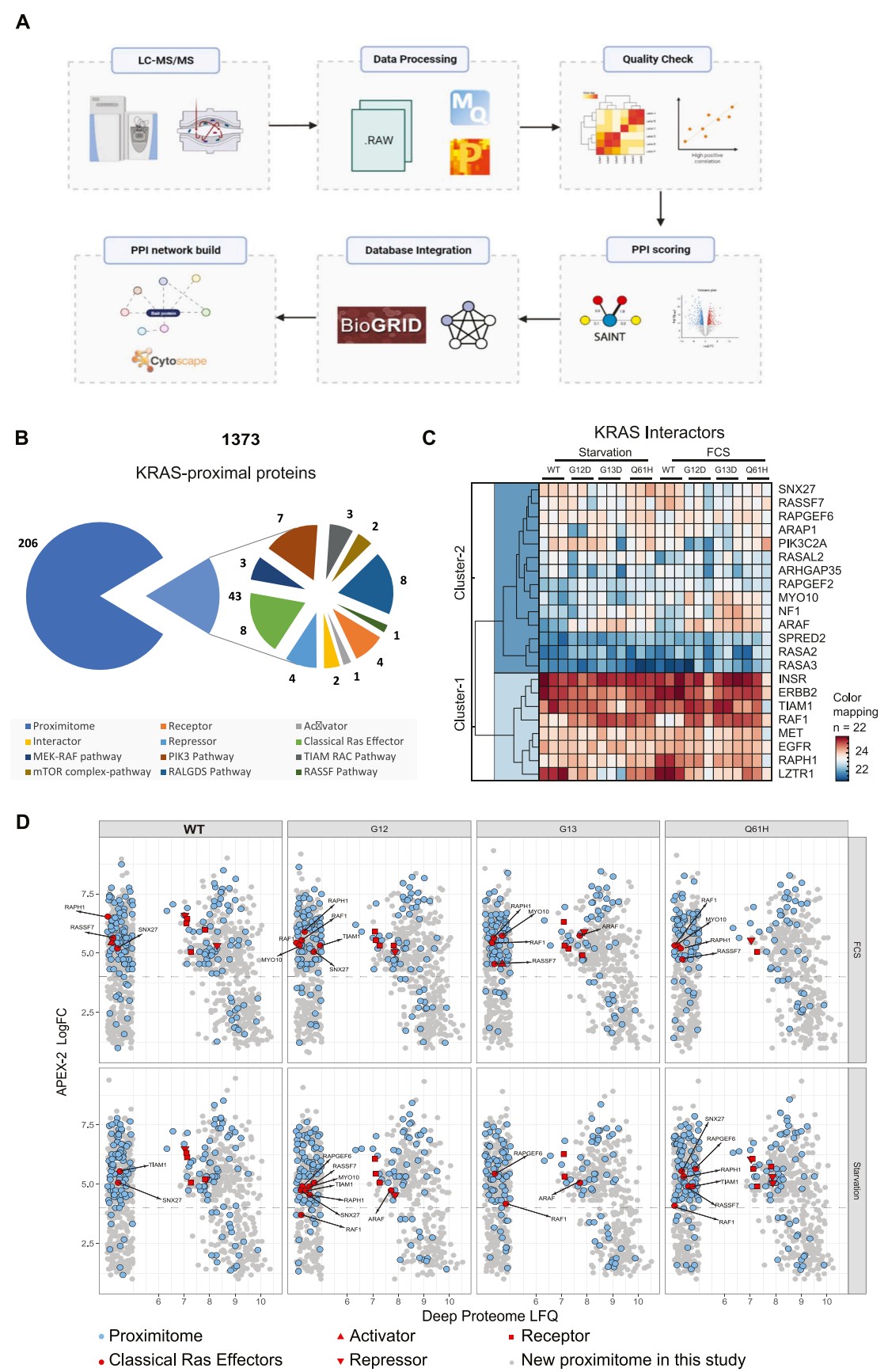

in our "proximitome" experiment showed no significant changes, except for LZTR1 in the comparison between WT and Q61H (Figs 4B and S3F and G, Table S2). We further integrated the proximity data with the proteomic data to gain a more comprehensive understanding of how protein abundance impacts the "proximitome" (Figs 4C and S3H and I, Table S2). ARAF and RAF1 were markedly enriched in the proximal environment of all KRAS mutants compared with WT KRAS without substantial changes in their protein levels (Figs 4C and S3H and I). Conversely, LZTR1 was notably increased in proximity to WT KRAS when compared with the G12D and G13D oncogenic mutants, with no significant alterations in its protein levels (Figs 4C and S3H), except in the case of Q61H KRAS mutant-expressing cells, where increased protein levels of LZTR1 were observed (Fig S3I). Furthermore, the proteomic data revealed a significant increase in mutated KRAS protein levels when compared with the WT, perhaps with the exception of the Q61H mutant (Figs 4B, S3F and G, and S4A–D). The observed changes in KRAS protein abundance could be attributed to varying KRAS half-lives between mutants and WT KRAS, a subject we decided to further explore. Collectively, these findings indicate unaltered LZTR1 protein abundance levels in cells expressing WT or KRAS mutants.

### Oncogenic KRAS mutants alter LZTR1-controlled KRAS protein turnover

Preferential enrichment of LZTR1 in the proximal proteome of WT KRAS relative to its mutants might be linked to protein turnover (Fig 4A). This was further supported by the proteomic data, where KRAS oncogenic mutants, especially G12D and G13D, showed a higher protein level as compared with WT KRAS (Figs 4B and S3F and G). To examine this hypothesis, we knocked down LZTR1 in HEK293 cells expressing WT KRAS-GFP, G12D-GFP, G13D-GFP, and Q61H-GFP fusion proteins. Knockdown of LZTR1 stabilises WT KRAS-GFP expression (Figs 4C and S5A and B), which is consistent with its role as an adaptor protein promoting ubiquitination and degradation of KRAS (Abe et al, 2020).

Notably, LZTR1 over-expression did not alter the stability of KRAS G12D and G13D KRAS mutants, perhaps with a slight effect on the Q61H mutant (Figs 4C, S3H and I, and S5A and B). This suggests that oncogenic KRAS mutants' half-life might be extended as compared with the WT counterpart. To test this, KRAS-GFP cell lines expressing WT and oncogenic KRAS mutants (G12D, G13D, and Q61H) were treated with cycloheximide (CHX) at different timepoints. WT KRAS was found to have a shorter half-life as compared with the oncogenic mutants (Fig 4D–F). To delineate the preferential affinity of LZTR1 towards WT KRAS as opposed to its mutant variants, we

orchestrated an over-expression of GFP-KRAS fusion proteins (WT and G12D) in the concurrent presence or absence of LZTR1 when monitoring both endogenous WT KRAS and exogenous GFP-KRAS. In scenarios where LZTR1 was knocked down, a significant up-regulation of both exogenous and endogenous WT KRAS was observed. Conversely, upon co-expression of WT KRAS and G12D KRAS within the same cellular milieu, the knockdown of LZTR1 predominantly stabilized endogenous WT KRAS but did not affect G12D protein levels (Fig S5C).

Based on this, we hypothesized that LZTR1 over-expression may preferentially destabilise WT KRAS over oncogenic mutants. Indeed, elevated LZTR1 levels resulted in substantially decreased WT KRAS protein levels, whereas the three oncogenic KRAS mutants showed minimal or no changes (Fig 4F). Interestingly, LZTR1 protein levels were highly up-regulated in the presence of oncogenic KRAS, suggesting a possible interdependence between KRAS and LZTR1 protein levels and turnover (Fig 4G). This was previously observed in our proteomic experiment, at least for the Q61H mutation, where LZTR1 protein levels were significantly higher in Q61H as compared with the WT (Fig S3G). To further explore how LZTR1 affected the protein stability of the oncogenic KRAS mutations, we measured protein levels of WT KRAS and mutants in the presence of LZTR1 using a CHX-chase assay (Fig 4G). In cells co-expressing Flag-KRAS WT and Myc-LZTR1, KRAS protein levels were reduced by ~50% after 1 h of exposure to CHX, whereas the three mutant protein levels remained unchanged (Fig 4G). To further explore differences between KRAS WT and mutated KRAS protein half-lives, we repeated the CHX experiment, extending the cell treatment duration to 10 h (Fig S5D and E). Interestingly, we observed that the presence of LZTR1 significantly impacted the stability of WT KRAS protein. In contrast, the effect was notably less pronounced in the case of the three oncogenic KRAS mutants. This suggests that LZTR1 exerts a considerable influence on the turnover of WT KRAS protein, whereas its impact is less pronounced for oncogenic KRAS mutants.

Moreover, we observed a notable difference in LZTR1 turnover as well. LZTR1 protein turnover appeared to be accelerated in the presence of WT KRAS as compared with the mutants (Fig S5D and E). This effect may not be directly associated with the mutation itself but rather with the levels of KRAS, suggesting that lower levels of KRAS could affect LZTR1 protein turnover (Fig S5F).

### WT KRAS and LZTR1/CUL3 regulatory circuit disrupted by oncogenic KRAS mutants

LZTR1 was described as a substrate adaptor of CUL3-based ubiquitin E3 ligase (Bigenzahn et al, 2018; Steklov et al, 2018). Therefore,

---

**Figure 2. KRAS WT, G12D, G13D, and Q61H proximal proteomes.**
**(A)** Mass spectrometry data analysis schematic representation. Data were initially analysed in MaxQuant to identify proteins present in our samples. Data were then processed with an interaction scoring algorithm SAINTexpress to provide a score on the probability of a true "interaction" (true proximity protein) with our bait. Finally, true proximity proteins were decided to include any protein with a SAINT score above 0.8 and fold difference higher or equal to four as compared with the beads control. On the other hand, MaxQuant results were further analysed on Perseus, and volcano plots were generated. Different databases, including BioGRID, were integrated to explore the identification of known interactors. Finally, PPI networks were generated through Metascape and Cytoscape. **(B)** All Ras proximal proteins are identified in each APEX2-KRAS "proximitome." SAINT probability ≥ 0.8; fold change ≥ 4 over beads control (1,373 proteins). Several KRAS interactome families (receptors, repressors, activators, interactors, classical Ras effectors, "proximitome," MEK-RAF pathway, PIK3 pathway, TIAM-RAC pathway, RALGDS pathway, RASSF pathway) were explored. **(C)** Heatmap showing the relative abundance of known KRAS interactors in WT KRAS, G12D, G13D, and Q61H KRAS mutant-APEX2 samples under either starvation or FCS-stimulated conditions. **(D)** Landscape (blue – previously known; grey – this study) of the KRAS-APEX2 "proximitome" (SAINT score ≥ 0.8) relative to the cellular proteome. Previously known Ras effectors, including receptors, repressors, and activators, are coloured red.

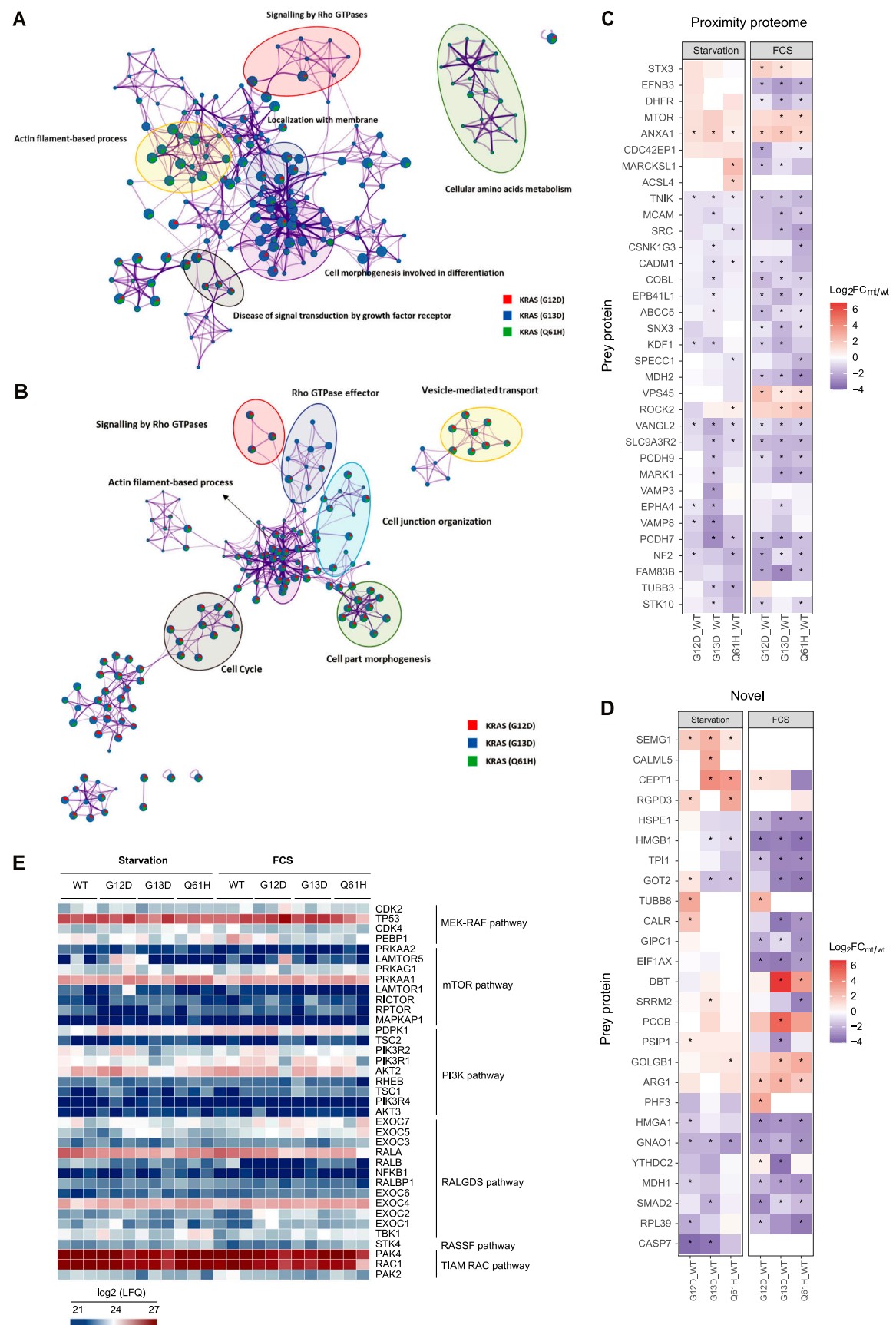

we examined whether inhibiting Cullin-RING E3 ubiquitin ligases (CRLs) via blocking Cullin neddylation affects KRAS and LZTR1 protein stability. MLN9424 is a small molecular inhibitor of the NEDD8-activating enzyme used to interfere with the function of CRLs (Milhollen et al, 2010). HEK293 cells were initially co-transfected with LZTR1 and WT or KRAS mutants, followed by treatment with different concentrations of MLN9424 for 24 h. Co-transfection of LZTR1 and WT KRAS reduced KRAS protein levels, which were restored in the presence of MLN9424 (Fig 4I). This was not observed with the oncogenic KRAS G12D and G13D mutants, perhaps to some extent with Q61H in the presence of high concentrations of MLN9424 (1 $\mu M$). Notably, LZTR1 protein levels were up-regulated with MLN9424 exposure, predominantly in the presence of WT KRAS. Together, LZTR1 and WT KRAS protein levels showed an interrelated abundance profile not observed with the mutants, perhaps except KRAS G12D and Q61H to a minor extent (Fig 4I). Furthermore, we observed that MLN9414 could restore LZTR1 protein levels, suggesting a potential regulation of LZTR1 protein levels by CULLINs. To study the underlying dynamics of this effect, MLN9424 inhibitor experiments were performed in a time-dependent fashion. HEK293T cells were co-transfected with Myc-LZTR1 and Flag-KRAS. Cells were then incubated with 0.3 $\mu M$ MLN9424 for 0, 6, and 18 h, respectively. WT KRAS protein levels were gradually increased, in contrast to the oncogenic KRAS mutants (Fig 4J). LZTR1 protein levels followed a comparable trend. Once more, LZTR1 protein levels were affected in an ML4294-dependent manner, predominantly in the presence of WT KRAS (Fig 4J). Because Cullin3 was previously shown to regulate (WT) KRAS protein levels via LZTR1 (Abe et al, 2020), we knocked down Cullin 3 to investigate how LZTR1 protein levels behaved in the presence of CHX. Removal of Cullin3 led to an increase in LZTR1's protein half-life, suggesting that Cullin 3 could regulate its co-factor (LZTR1) (Fig S6A). To further examine how KRAS protein levels affect LZTR1 protein stability, HEK293 cells were co-transfected with LZTR1 and KRAS, where LZTR1 plasmid concentration was kept constant, whereas KRAS concentration was varied (Fig S6B). We observed that LZTR1 protein levels were stabilised in a dose-dependent similar to KRAS, suggesting a direct crosstalk between KRAS and LZTR1. This appeared to be Cullin3-dependent because when Cullin 3 was knocked down, Flag-KRAS and Myc-LZTR1 co-expression did not lead to LZTR1 accumulation whereas Flag-KRAS protein levels were stabilised (Fig S6C). To exclude that alterations in LZTR1 protein levels in conjunction with mutant KRAS were attributable to increased mRNA levels, we used a FLP T-REX GFP-stable KRAS cell line system. We induced over-expression of WT or the three oncogenic KRAS mutants and used RT–qPCR to monitor the LZTR1 mRNA levels that were found unchanged, suggesting that any observed changes are not a consequence of mRNA elevation (Fig S6D). Altogether, we conclude that the regulation of KRAS and LZTR1 turnover is linked, possibly involving Cullin-Ring Ubiquitin Ligases.

# Discussion

Oncogenic KRAS G12D, G13D, and Q61H are among the most common driver mutations found in human cancer (Liu et al, 2019; Cook et al, 2021). KRAS mutations occur in different cell types with allele-specific profiles, suggesting dissimilar underlying mechanisms behind driving cell transformation. For instance, G12D KRAS mutations occur in over 90% of pancreatic ductal adenocarcinoma (Wong et al, 2016); G13D is most frequent in colon adenocarcinoma (Taieb et al, 2017); and Q61H in lung adenocarcinoma (Kunimasa et al, 2020). KRAS-dependent oncogenesis depends on at least four factors contributing to different KRAS mutations associated with specific cancer types (Prior et al, 2012). First, expression levels and GTP/GDP-KRAS ratios contribute to cell transformation. Second, active KRAS dynamics vary considerably between oncogenic mutations (Killoran & Smith, 2019). Third, KRAS mutations may alter GEF, GAF, and effector interactions (Kiel et al, 2021). Finally, the genetic, epigenetic, and proteomic cell type-specific environment may differentially impact RAS-dependent oncogenesis.

Here, we systematically investigated how these oncogenic KRAS driver mutations affect their proximal molecular environment. Using an isogenic cell line as a control reduced interference because of different cellular backgrounds. Our approach relied on an over-expression system, which is known to potentially provoke cell transformation (Ito et al, 2021). However, we have carefully controlled this with an optimised starvation protocol to mimic a "baseline state." We cannot exclude the possibility of missing crucial KRAS associations in our experimental system because of the intrinsic limitations of over-expression. An APEX2 proximity labelling method was used as it functions at a timescale of seconds, which is advantageous for capturing transient "interactions" and fast molecular changes as compared with BioID-based approaches (Trinkle-Mulcahy, 2019). Labelling conditions were optimised, and levels of protein biotinylation were assessed in whole cell lysates and streptavidin-evaluated material to control for efficacy of capture (Fig 1D). Previous KRAS "proximitome" studies based on BioID revealed many shared proteins between WT and mutant KRAS (mainly G12V, S17N, C185S), predominantly involved in protein expression and trafficking (Adhikari & Counter, 2018). In addition, there appears to be a significant overlap between GTP/GDP-KRAS proteins detected in close proximity, indicating a more gradual KRAS activation transition rather than an on/off switch. Very few specific interactors were reported, such as the KRAS G12V

---

**Figure 3. Enriched proteomes differ between KRAS WT and G12D, G13D, and Q61H mutants.**
**(A, B)** Metascape enrichment network visualisation showing the intra-cluster and inter-cluster similarities of enriched terms (up to 10 terms per cluster) in starvation (A) and FCS stimulation state (B) separately, where nodes are represented by pie charts indicating their associations with three different mutants. The enriched terms with a Kappa-test-based similarity score > 0.3 are linked by an edge (the thickness represents the similarity score). The network is visualised with Cytoscape (v3.1.2) with a "force-directed" layout and with edge bundled for clarity. The colour code for the pie sector represents G12D (red), G13D (blue), and Q61H (green). Cluster labels were added manually. **(C, D)** Heatmap showing the $\log_2$ (FC MUT/WT) of differentially enriched proximal proteins, which were reported in previous "proximitome" studies (C) or proposed based on our study. **(D)** Asterisk indicates that the hit meets the criteria of $\log_2$ (mutant/WT FC) > 0.5 or <–0.5 and $–\log_{10}$ ($P$-value) > 0.7. **(E)** Heatmap showing the relative abundance ($\log_2$ LFQ) of proximal proteins involved in KRAS-related canonical pathways (such as MEK-RAF, mTOR, PI3K, RALGDS, RASSF, and TIAM-RAC pathway).

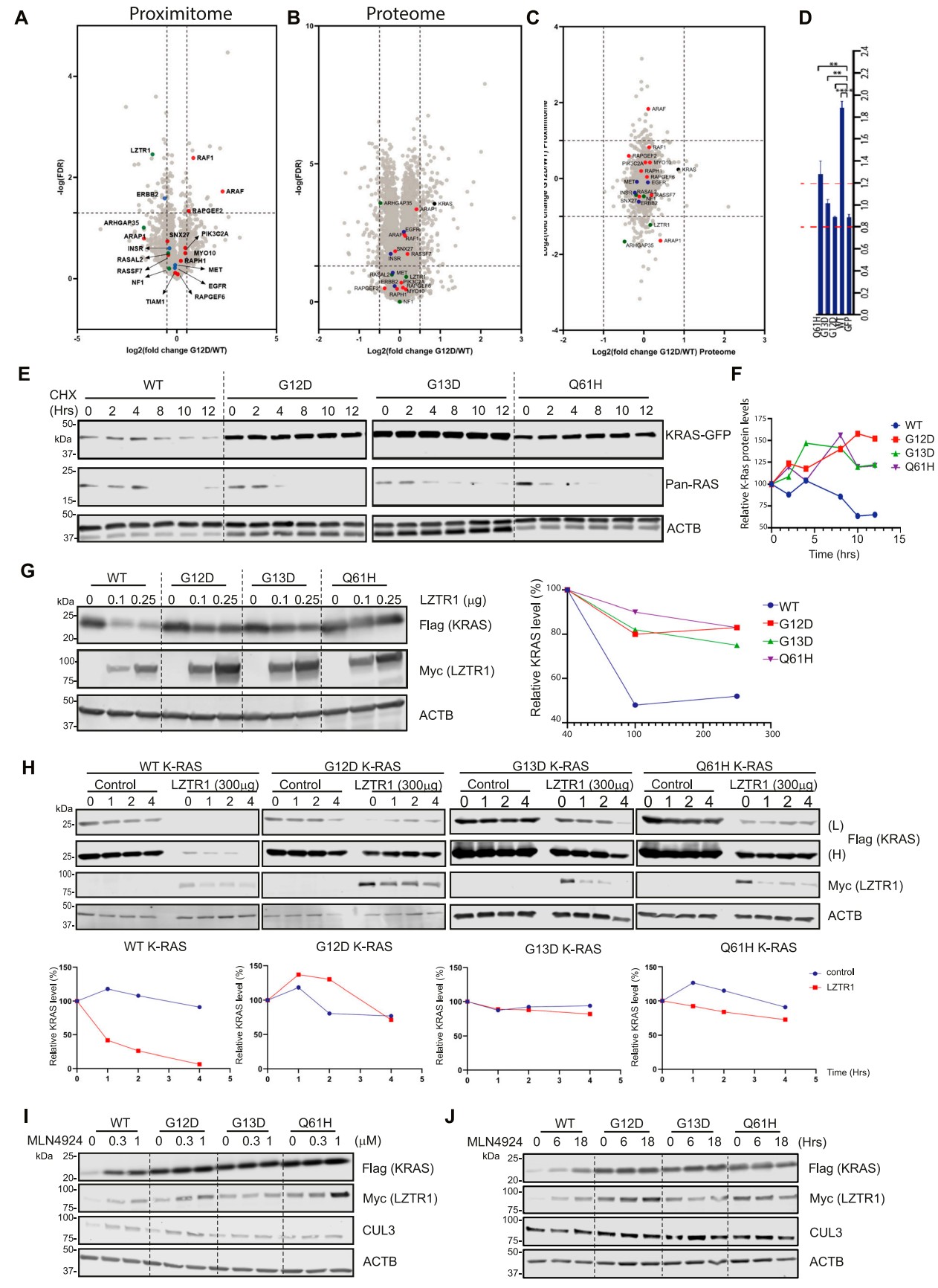

selective interaction with phosphatidylinositol-4-phosphate 5-kinase type 1 (PIP5K1), mediating PI3K and ERK signalling (Adhikari & Counter, 2018).

Our approach offers deeper insights as APEX2 may capture more concise molecular snapshots because of shorter labelling times. We systematically compared the molecular environment between WT KRAS and G12D, G13D, and Q61H mutants under resting and acute FCS stimulation conditions. Residues G12, G13, and Q61 are located within the KRAS G domain, altering nucleotide binding and exchange, and are expected to alter KRAS interacting networks. To further examine proteins involved in known KRAS signalling pathways, such as RAF-MEK, mTOR, PI3K, RALGDS, RASSF, and TIAM-RAC networks, they were selected and visualised (Fig 3E). 37 downstream signalling proteins were identified, for which no significant differences were noted between WT KRAS and all the mutants. Gene ontology-cellular component analysis for WT and the mutants showed enrichment of focal adhesion and plasma membrane, in line with its subcellular localisation (Fig S1), consistent with previous studies (Kiel et al, 2021).

The absence of some canonical KRAS interactors, such as BRAF, HRAS/NRAS, and SOS, could be explained by the lack of accessible lysine residues for the biotinylation reaction to take place or perhaps their low expression level in HEK293 cells as it is the case for b-Raf (Geiger et al, 2012). ARAF, in turn, seems to be strongly enriched with KRAS G13D as compared with G12D and Q61H. On the other hand, NF1 and SPRED2 seem to be enriched with the G13D and Q61H mutants as compared with G12D, in line with previous findings that the G12D KRAS signalling pathway is not affected by NF1 (Cheng et al, 2021). Both ARAF and RAF1 demonstrated enrichment within the microenvironment of mutant KRAS, with no accompanying alterations in proteomic levels. This suggests that the observed protein enrichment within the "proximitome" indicates elevated association rather than cellular abundance.

As shown previously (Bigenzahn et al, 2018; Abe et al, 2020), LZTR1 was detected in the microenvironment of both WT and selected KRAS mutants. Most notably, we found that LZTR1 levels were strongly reduced in the proximity of the mutants G12D and G13D and, to a lesser extent, with Q61H (Fig 4A). We subsequently demonstrated that cellular LZTR1 mRNA or protein levels do not contribute to this phenotype, strongly suggesting that LZTR1 is significantly more enriched in the WT KRAS microenvironment as compared with the mutants. LZTR1's effects on ubiquitination and turnover of WT KRAS and some mutants have been examined (Abe et al, 2020). Here, we extended this observation to the KRAS mutants G12D, G13D, and Q61H, whose turnover was less dependent on LZTR1 (Fig 4C, F, and G).

The protein half-life of mutated KRAS used in this study appears to be dramatically increased when compared with WT KRAS (Fig 4E). Furthermore, the effect of LZTR1 on the protein stability of KRAS is significantly attenuated by its oncogenic mutations (Figs 4H and S5D and E). The observed difference could potentially stem from a direct effect of the interaction between KRAS and LZTR1. This interaction might be influenced by the oncogenic mutations in KRAS, resulting in altered stability or turnover dynamics of the KRAS protein. In addition, intracellular factors within the cellular microenvironment surrounding the KRAS proteins could also contribute to these alterations, potentially affecting cellular processes that modulate oncogenic KRAS mutation protein levels. LZTR1 was increased more in the "proximitome" of WT KRAS than the mutated KRAS, which could explain why oncogenic KRAS proteins are more stable. However, it is unclear whether this is because of less physical interaction or reduced accessibility of LZTR1 to the KRAS protein. Further investigation into these possibilities could provide valuable insights into the observed differences. LZTR1 protein levels may be regulated by Cullins, thereby affecting KRAS protein levels, especially in the case of Q61H. This is consistent with our data in the context of ectopic expression and measuring endogenous levels of LZTR1 by proteomic analysis (Figs 4B and S3F and G).

Small-molecule KRAS inhibitors (KRASi) may perturb this effect (Kim et al, 2023). Also, changes in proteostasis reprogramming of the KRAs microenvironment may contribute to KRASi resistance (Lv et al, 2023) Alternatively, LZTR1 mutations in certain Rasopathies, such as glioblastoma and dominant Noonan syndrome, may also disrupt KRAS turnover (Steklov et al, 2018; Motta et al, 2019). Therefore, because the regulation of KRAS protein levels by LZTR1 is prevented by oncogenic mutations, it is worth exploring its implications for malignant cell transformation and cancer.

**Figure 4. Oncogenic mutations of KRAS modulate its turnover by LZTR1/CUL3.**
**(A, B)** Volcano plots visualizing molecular differences between G12D and WT KRAS under starvation conditions. **(A, B)** Fold changes and *P*-values are shown resulting from a *t* test for proximal enriched proteins (A) and the total proteome (B) identified in the G12D KRAS mutant under starvation-treated conditions, with WT KRAS used as a control. Known KRAS effectors are highlighted in red, repressors in green, receptors in blue, and KRAS in black. **(C)** A scatter plot comparing the KRAS WT versus G12D mutant "proximitome" and proteome. The Y-axis represents fold changes between the G12D mutant and the WT "proximitomes." The X-axis reflects fold changes between the G12D mutant and WT proteomes. Known KRAS effectors are highlighted in red, repressors in green, receptors in blue, and KRAS in black. **(D)** Cells were seeded in 96-well plates and treated with 1 μg/ml tetracycline for 24 h. The next day, cells were transfected with LZTR1 siRNA or control siRNA. Cells were imaged after 72 h using the Opera Phenix microscope. The images were analysed in the Columbus Image Data Storage and Analysis System. The median fluorescence per well per mean per cell was determined. Finally, a fold difference was measured between the control and the siRNA LZTR1. **(E)** Stable HEK293 FRT T-REX cell lines of WT, G12D, G13D, and Q61H were treated with 1 μg/ml tetracycline for 24 h. The next day, cells were treated with 50 μg/ml cycloheximide (CHX). At the indicated time points after CHX treatment, cells were harvested. Western blot analysis showed GFP, Pan-Ras, and ACTB in-cell extract. **(F)** Relative intensities from Fig 4D were analysed using ImageStudio, and expression was normalised to ACTB protein levels to determine KRAS half-life. This series of experiments helps elucidate the relationship between KRAS and LZTR1 protein levels and turnover, showing how different KRAS mutations disrupt this interaction. **(G)** HEK293 cells were seeded in six-well plates and transfected with 1 μg Flag-K-Ras plasmids and 0, 0.1, or 0.25 μg Myc-LZTR1 plasmid concentration. Cells were cultured for a total of 48 h before protein levels were evaluated using Western blot. Relative intensities were analysed using ImageStudio, and expression levels were normalised to vector control. **(H)** HEK293T cells were transfected with 1 μg Flag-RAS and 0.2 μg Myc-LZTR1 plasmid concentration. Cells were incubated for 48 h, treated with 50 μg/ml CHX, and harvested at the indicated time points. Western blot analysis of FLAG (RAS), MYC (LZTR1), and ACTB in-cell extract is shown. Band densities were analysed using GelAnalyzer software, and expression was normalised to ACTB levels. **(I)** HEK293T cells were transfected with 1 μg Flag-RAS and 0.2 μg Myc-LZTR1 plasmid and allowed to grow for 24 h with MLN4924 using different concentrations (0, 0.3, 1 μM, respectively). Cell extract was then analysed by Western blotting, and FLAG (RAS), MYC (LZTR1), CUL3, and ACTB were monitored. **(J)** HEK293T cells were transfected with 1 μg Flag-RAS and 0.2 μg Myc-LZTR1 plasmid concentrations and allowed to grow for 48 h. Cells were then treated for the indicated times with MLN4924 (0.3 μM). Cell extracts were then analysed on Western blot, and FLAG (RAS), MYC (LZTR1), CUL3, and ACTB were monitored.

# Materials and Methods

## Cell lines

HEK293T and Flp-In T-REx 293 Cell Line (parental line; R78007; Thermo Fisher Scientific) were cultured in DMEM (high glucose, #12440-53; Gibco) supplemented with 10% FBS (#10500-64; Gibco), 2 mM L-glutamine (G7513; Sigma-Aldrich), 100 $\mu$g/ml Zeocin (ant-zn-1; InvivoGen), 15 $\mu$g/ml blasticidin (ant-bl-1; InvivoGen) at 37°C in a humidified 5% $CO_2$ atmosphere. The stable transfected Flp-In T-REx 293 cell lines (KRAS WT-APEX2, KRAS G12D-APEX2, KRAS G13D-APEX2, KRAS Q61H-APEX2, KRAS WT-GFP, KRAS G12D-GFP, KRAS G13D-GFP, KRAS Q61H-GFP, GFP only) were cultured in the same medium and selected by adding 100 $\mu$g/ml of hygromycin B (ant-hg-1; InvivoGen) to the media. For the induction of the stable transfected Flp-In T-REx 293 cell lines, 1 $\mu$g/ml tetracycline was added to the media.

## Plasmids

Q5 High-Fidelity DNA Polymerase (M0491S; NEB) was used to perform PCR reactions following the manufacturer's instructions. APEX2 plasmid was donated from Pedro Carvalho's laboratory (pcDNA3 APEX2-NES). APEX2 ORF was then PCR amplified and inserted into the pcDNA5/FRT vector (Invitrogen) using the NEBuilder HiFi DNA Assembly (E5520S; NEB) following the manufacturer's protocol. KRAS ORF was initially codon optimised and synthesised from Eurofins Scientific. Then KRAS ORF was PCR amplified and cloned into the pcDNA5/FRT Vector APEX2 plasmid using NEBuilder HiFi DNA Assembly following protocol. Different KRAS mutants were generated using Q5 Site-Directed Mutagenesis Kit (E0554S; NEB). The successful clones were confirmed using DNA sequencing.

A pcDNA5/FRT/TO GFP (19444; ITEM) was purchased from Addgene. PCR was used to amplify all the different KRAS (WT, G12D, G13D, Q61H) ORF and the pcDNA5/FRT/TO GFP. NEBuilder HiFi DNA Assembly was used to generate the GFP-tagged plasmids. The successful plasmids were confirmed by DNA sequencing. The pcDNA3.1+ Flag-KRAS-G12C plasmid was a kind gift from Dr. Vikram Rao (Pfizer). WT, G12D, G13D, and Q61H KRAS were generated using the Q5 Site-Directed Mutagenesis Kit. The successful clones were confirmed using DNA sequencing. LZTR1 ORF and the pCI-backbone (41552; Addgene) were PCR amplified and cloned using NEBuilder HiFi DNA Assembly.

## Immunoblotting

5 × 10$^6$ HEK293T or HEK293 FRT were washed with ice-cold PBS and lysed with RIPA buffer containing protease and phosphatase inhibitors. For immunoblotting, 25 $\mu$g of protein were then fractionated on Tris–glycine SDS–PAGE gradient (4–15% acrylamide) gels (#3450123; Bio-Rad), transferred onto nitrocellulose membranes (Millipore), and detected with the indicated antibodies using an LI-COR detection system. Primary antibody used in this study: DYKDDDDK Tag Antibody (Flag, 2368; Cell Signalling Technology), A-Raf Antibody (4432; Cell Signalling Technology), LZTR1 (sc-390166; Santa Cruz Biotechnology), p44/42 MAPK (Erk1/2) (9107;

Cell Signalling Technology), Phospho-p44/42 MAPK (Erk1/2) (Thr202/Tyr204) (4370; Cell Signalling Technology), GAPDH (G8795; Sigma-Aldrich).

## Generation of stable cell lines

Flp-In T-REx 293 Cell Line (Invitrogen) was transfected using Lipofectamine LTX following the manufacturer's protocol. 300 $\mu$g/ml of hygromycin B was added the next day, and cells were incubated for two weeks when the medium was changed every four days. Finally, the successful population of cells was further used for the described experiments.

## Proximity labelling

5 × 10$^6$ HEK293 FRT were induced with 1 $\mu$g/ml tetracycline overnight. The day after, cells were incubated with 500 $\mu$M of biotinyl tyramide for 30 min. A final concentration of 1 mM $H_2O_2$ was added for 40 s. The cells were washed two times with quencher buffer (10 mM sodium ascorbate, 10 mM sodium azide, 5 mM Trolox), two times with ice-cold PBS, and finally with quencher buffer for a last time. The cells were lysed with RIPA buffer containing 1 mM PMSF, 5 mM Trolox, 10 mM sodium azide, 10 mM sodium ascorbate, protease, and phosphatase inhibitors.

## Transient transfection

For DNA plasmids, 5 × 10$^6$ HEK293T cells were grown in six-well plates, and Lipofectamine 3000 Transfection Reagent (L3000001; Thermo Fisher Scientific) was used to transfect the cells following the manufacturer's instructions. The concentration of plasmids used was indicated in each experiment.

For siRNA transfection experiments, HEK293T cells were grown in six-well plates, and RNAimax transfection reagent (#13778-150; Invitrogen) was used following the manufacturer's protocol. A final concentration of 10 nM of the following siRNAs was used: LZTR1 siRNA on-TARGET plus SMARTpool (L-012318-00-0005; Dharmacon) and CUL3 siRNA on-TARGET plus SMARTpool (L-010224-00-0005; Dharmacon).

## Cycloheximide chase experiment

5 × 10$^6$ HEK293T and HEK293 FRT were transfected with the indicated plasmids for 24 h. A fresh medium was added for an additional 24 h, and then 50 $\mu$g/ml cycloheximide (CHX) was added. Western blot was used to evaluate different expressions of proteins.

## RAS activation kit

5 × 10$^6$ HEK293 FRT were incubated with or without FCS for the indicated time. Cells were washed with ice-cold PBS and lysed with the designated lysis buffer from the active Ras detection kit provided. Subsequent steps were performed as the manufacturer's protocol for the active Ras detection kit (8821; Cell Signaling Technology).

### Live cell imaging

10,000 HEK293 FRT were seeded in 96-well plates (M33089; Thermo Fisher Scientific) and induced with tetracycline for 24 h. The next day, cells were transfected with LZTR1 siRNA or control siRNA. 72 h later, we removed the medium. Hoechst 33342 (H3570; Thermo Fisher Scientific) was diluted in PBS in 1:2,000 dilution and added to the cells. The cells were then incubated for 10 min in the incubator. Cells were imaged using the Opera Phenix microscope. Images were further analysed in the Columbus Image Data Storage and Analysis System. The media cell fluorescent per mean fluorescent per well was then determined for each condition. Finally, the siRNA control was used to normalise the data. The experiment was performed in biological triplicate, and the graph was generated using Prism software (GraphPad Prism 9.2.0).

### Sucrose gradient fractionation

HEK293 FRT cells were homogenised in isotonic buffer (0.25 M sucrose, 10 mM Tris–HCl, pH 7.5, 10 mM KCl, 1.5 mM MgCl2, and protease inhibitor cocktail). After homogenisation, whole cell lysates were centrifuged at 10,000$g$ for 10 min to obtain the crude membranes. P10 crude membranes were washed once with the same isotonic buffer and resuspended in the same buffer. 60% and 20% sucrose solutions were prepared in low salt buffer (10 mM Tris–HCl, pH 7.5, 10 mM KCl, 1.5 mM MgCl2), and a continuous sucrose gradient was generated using BIOCOMP Model 108 GRAGIENT MASTER according to the manufacturer's guidance. P10 crude membranes were loaded on top of the gradient and centrifuged at 170,000$g$ for 2 h 12 fractions (1 ml per fraction) were collected from top to bottom.

### Membrane-cytoplasmic cellular fractionation

Fresh HEK293 FRT cell pellets were suspended in digitonin lysis buffer (50 mM Hepes pH 7.5, 10 mg/ml digitonin, 150 mM NaCl) and incubated for 30 min at 4°C under spinning. The sample was spun at 6,000$g$ for 5 min at 4°C. The supernatant (cytoplasmic fraction) was transferred to a new tube. The pellet was resuspended in 0.3% NP-40, 50 mM Hepes pH 7.5, and 150 mM NaCl and incubated on ice for 5 min. The samples were centrifuged at 1,500$g$ for 5 min. The supernatant (membrane fraction) was transferred to a new tube and stored at –20°C until analysis.

### Streptavidin immunoprecipitation

For immunoprecipitation (IP) experiments, Dynabeads MyOne Streptavidin T1 was used, 30 $\mu$l for Western blotting and 110 $\mu$l for mass spectrometry experiments. Initially, the beads were washed twice with RIPA buffer, once with 1 M KCL, 0.1 M $Na_2CO_3$, and 2 M Urea in 10 mM Tris–HCL, pH 8, and, finally, twice with RIPA buffer again. For Western blot analysis, biotinylated proteins were eluted using 60 $\mu$l of 2X loading buffer supplemented with 2 mM biotin and 20 mM DTT by boiling (98°C) for 10 min. For Mass spectrometry analysis, an on bead-based digestion protocol was used as described below.

### On-beads digestion and mass spectrometry sample preparation

Immunoprecipitated protein samples were denatured in 8 M urea in 100 mM triethylammonium bicarbonate buffer (TEAB) for 30 min at room temperature. Afterwards, the samples were reduced with 10 mM tris(2-carboxyethyl)phosphine (TCEP) for 30 min at room temperature. 50 mM iodoacetamide was added to the sample for 30 min at room temperature to alkylate the proteins. Samples were then diluted down to 1.5 M urea with 50 mM TEAB. Finally, 250 ng trypsin (Promega) dissolved in XmL 50 mM TEAB was added, and samples were incubated at 37°C overnight. The day after, samples were desalted using a C18 solid-phase cartridge (Sep-Pak; Waters) following the manufacturer's protocol. Purified peptide eluates were dried by vacuum centrifugation, resuspended in buffer A (98% MilliQ-$H_2$0, 2% CH3CN, and 0.1% TFA), and stored at –20°C until analysis.

### Liquid chromatography-tandem mass spectrometry (LC–MS/MS) analysis

For proximity labelling, LC–MS/MS analysis was performed as previously described (Fischer & Kessler, 2015). In brief, resuspended peptide material was trapped on an AcclaimPepMap 100 C18 HPLC Column (PepMapC18; 300 $\mu$m × 5 mm, 5 $\mu$m particle size; Thermo Fisher Scientific) using solvent A (98% MilliQ-$H_2$0, 2% CH3CN, and 0.1% TFA) at a flow rate of 8 $\mu$l/min and separated on an Ultimate 3000UHPLC system (Thermo Fisher Scientific) coupled to a Q-Exactive mass spectrometer (Thermo Fisher Scientific). The peptides were separated on an EASY-Spray PepMap RSLC column (75 $\mu$m i.d. × 2 $\mu$m × 50 mm, 100 Å, 250 nl/min; Thermo Fisher Scientific) using a linear gradient (length: 60 min, 5–35% of acetonitrile in 0.1% formic acid/5% DMSO). The raw data were acquired on the mass spectrometer in a data-dependent mode. Full scan MS spectra were acquired with a resolution of 70,000 at 200 m/z (scan range 380–1,800 m/z, AGC target 3 × 10$^6$, maximum injection time 100 ms). The 15 most intense peaks were selected for HCD fragmentation at 28% of normalised collision energy. HCD spectra were acquired at a resolution of 17,500 (AGC target 1 × 10$^5$, maximum injection time 128 ms) with the first fixed mass at 100 m/z.

For total proteome analysis, LC–MS/MS was performed using the Thermo Fisher Scientific Vanquish Neo UHPLC system connected to a Thermo Orbitrap Ascend mass spectrometer. The Vanquish Neo was operated in the "Trap and Elute" mode using a PepMap Neo trap (185 $\mu$m, 300 $\mu$m × 5 mm) and EASY-Spray PepMapNeo column (50 cm × 75 $\mu$m, 1500 bar). 1.5% of the tryptic peptides were trapped and separated using a 60-min linear gradient over 60 min (from 3% to 20% B in 40 min and from 20 to 35% in 20 min) at 300 nl/min flow. MS data were acquired in data-independent mode (DIA) with some modifications as compared with the previously described method (Muntel et al, 2019; O'Brien et al, 2023). Survey scans (MS1) were acquired in the Orbitrap over the mass range of 350–1,650 m/z, with a resolution of 45K resolution, maximum injection time of 91 ms, an AGC set to 125%, and an RF lens set at 30%. MS2 scans were then collected using the tMSn scan function, with 40 predefined variable-width DIA scan windows (Muntel et al, 2019) with a 30 K orbitrap resolution, normalized AGC target of 1,000%, maximum

injection time set to auto, and a 30% collision energy. Raw mass spectrometry files were quantified using a label-free approach with DIA-NN software (version 1.8).

## Proteomics data analysis

For proximity labelling, raw mass spectrometry (MS) data sets were processed using MaxQuant software (v1.6.14.0) and searched against human protein sequences in UniProt/KB (version: 2019). For proteome analysis, raw mass spectrometry (MS) data sets were processed using DIAN-NN 1.8.1 software and searched against human protein sequences in UniProt/KB (version: 2019). Perseus (version 1.6.15.0) and Significance Analysis of Interactome (SAINT) workflows 36 were applied for data analysis. The MS data generated in this study have been submitted to the PRIDE public repository with the accession number PXD029725.

## Data analysis

We characterised differentially enriched proteins in G12, G13, and Q61H mutants versus WT during starvation and FCS. We first used Perseus software (version 1.6.15.0) to perform a $t$ test to evaluate log fold changes ($Log_2$ FC) between WT and mutants. For each condition, we then carried forward the differentially expressed proteins with a $log_2$ (mutant/WT FC) > 0.5 or <−0.5 and −$log_{10}$ (q value) > 1.3. Proteins that fulfilled the above filtering criteria were considered potential mutant- or WT KRAS-specific associating proteins. Gene ontology enrichment analysis of differential proximal proteins for three mutants was performed using the Metascape web-based tool (www.metascape.org) (Zhou et al, 2019). To further investigate whether KRAS mutants G12D, G13D, and Q61H differ in downstream signalling pathways and known interactors, we compared the protein components in different pathways (MEK-RAF, mTOR, PI3K, RALGDS, RASSF, and TIAM-RAC) and known KRAS interacting partners enriched with different mutants and WT KRAS. Protein abundance was shown as a $Log_2$ (LFQ) value in the heatmap shown in Fig 3E.

We also investigated the overlap between our significantly enriched proteins ($log_2$ [mutant/WT FC] > 0.5 or <−0.5 and −log10 [q value] > 0.8) and various gene sets, including (i) GO molecular function, (ii) GO biological processes, (iii) GO cellular compartments, (iv) MSigDB Hallmark genes, and (v) CORUM complexes (Giurgiu et al, 2019). To do this, we used Genoppi (Pintacuda et al, 2021) to perform a one-tailed hypergeometric overlap test to assess the probability of finding a given gene set among our significantly enriched proteins.

## Quantitative real-time PCR (RT-qPCR)

HEK293 cells, stably expressing WT KRAS or mutant KRAS G12D, G13D, and Q61H, were plated at $0.5 \times 10^6$ cells per well in a six-well plate. After being treated or not with tetracycline for 24 h, cells were lysed in 350 $\mu$l of Buffer RLT (QIAGEN) per six-well. Total RNA was extracted using the RNeasy Mini Kit (QIAGEN) following the manufacturer's instructions. RNA was eluted with 30 $\mu$l of RNase-free water. Complementary DNA (cDNA) was synthesised from 1 $\mu$g of total RNA per condition using MultiScribe Reverse Transcriptase

(Applied Biosystems). RT-qPCR was performed using the LightCycler 480 SYBR Green I Master Kit and LightCycler 480 PCR System (Roche) according to the manufacturer's instructions. The Delta-Ct method was used to calculate the relative RNA expression levels. Primers used in this study are as followed: LZTR1-For: 5′-AGCGTGGACTTC-GACCATAG-3′; LZTR1-Rev: 5′-GCCAGCGATGCACTGTTTC-3′; KRAS-For: 5′-ACAGAGAGTGGAGGATGCTTT-3′; KRAS-Rev: 5′-TTTCACA-CAGCCAGGAGTCTT-3′; ACTB-For: 5′-CATGTACGTTGCTATCCAGGC-3′; ACTB-Rev: 5′-CTCCTTAATGTCACGCACGAT-3′.

## Software developed for data analytics

All analyses were performed with R version 4.0.2. Workflows and scripts have been compiled in a custom R package that can be found in the following repository: https://github.com/frhl/KRAS-methods.

# Supplementary Information

# Acknowledgements

We thank members of the Kessler group and ITEN teams for helpful comments and insightful discussions. This work was supported by Pfizer. Work in the BMK lab was funded by the Chinese Academy of Medical Sciences (CAMS) Innovation Fund for Medical Science (CIFMS), China (grant number: 2018-I2M-2-002) and an EPSRC grant EP/N034295/1.

## Author Contributions

A Damianou: conceptualization, data curation, formal analysis, supervision, investigation, visualization, methodology, and writing—original draft, review, and editing.
Z Liang: conceptualization, data curation, formal analysis, investigation, visualization, methodology, and writing—original draft, review, and editing.
F Lassen: software and visualization.
I Vendrell: data curation and writing—review and editing.
G Vere: data curation and software.
S Hester: data curation.
PD Charles: resources and software.
A Pinto-Fernandez: conceptualization and investigation.
A Santos: conceptualization and investigation.
R Fischer: investigation.
BM Kessler: conceptualization, supervision, funding acquisition, project administration, and writing—original draft, review, and editing.

## Conflict of Interest Statement

The authors declare that they have no conflict of interest.

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
