## [Reviewer comments · Life Science Alliance]

Life Science Alliance

Oncogenic Mutations of KRAS Modulate Its Turnover by the CUL3/LZTR1 E3 ligase Complex

Andreas Damianou, zhu liang, Frederik Lassen, Iolanda Vendrell, George Vere, Svenja Hester, Philip Charles, Adan Pinto-Fernandez, Alberto Santos, Roman Fischer, and Benedikt Kessler

DOI: <https://doi.org/10.26508/lsa.202302245>

Corresponding author(s): *Andreas Damianou, University of Oxford and Benedikt Kessler, Target Discovery Institute*

Review Timeline:

Submission Date:	2023-06-30
Editorial Decision:	2023-08-17
Revision Received:	2023-12-22
Editorial Decision:	2024-01-25
Revision Received:	2024-02-27
Accepted:	2024-02-27

Scientific Editor: *Eric Sawey, PhD*

Transaction Report:

August 17, 2023

Re: Life Science Alliance manuscript #LSA-2023-02245-T

Dr. Andreas Damianou
University of Oxford
United Kingdom

Dear Dr. Damianou,

Thank you for submitting your manuscript entitled "Oncogenic Mutations of KRAS Modulate Its Turnover by the CUL3/LZTR1 E3 ligase Complex" to Life Science Alliance. The manuscript was assessed by expert reviewers, whose comments are appended to this letter. We invite you to submit a revised manuscript addressing the Reviewer comments.

Thank you for this interesting contribution to Life Science Alliance. We are looking forward to receiving your revised manuscript.

Sincerely,

B. MANUSCRIPT ORGANIZATION AND FORMATTING:

Reviewer #1 (Comments to the Authors (Required)):

Damianou et al. utilized the APEX-based approach to identify the proximitome of wild-type KRAS and several KRAS mutants. Even though multiple TAP-based and BioID studies focused on the characterization of RAS interactome have been published, the APEX approach allows rapid proximity labeling and could reveal dynamic associations of active and inactive forms of KRAS in response to serum stimulation. Nonetheless, the observed differences between starved and serum stimulation conditions are limited. The ubiquitin ligase LZTR1 is the only protein that showed a significant difference in the interaction between wt and mutant KRAS. In line with the interaction data, the authors showed that LZTR1 affects the stability of wt but mutant KRAS. They also proposed a feedback regulation between wt-KRAS and LZTR1 protein levels. Even though these results suggest a novel mechanism of mutant KRAS activation due to disrupted LZTR1-mediated degradation, the manuscript has some major limitations that should be addressed by the authors:

1. Fig. 4B - is problematic. pERK levels are lower upon serum stimulation of the control samples. There is no control for RAS-Apex overexpression levels that could affect the interpretation of the results.
2. Fig. 4A,B - Why did serum stimulation leading to the activation of KRAS increase the interaction with LZTR1, whereas activating RAS mutations inhibit the binding?
3. Fig. 4C -The setup of this experiment does not make sense as the observed difference could be explained by the difference in the expression levels but not degradation. The experiment should be done like it is described in PMID: 18988848, which is considered a standard approach in the field. LZTR1 depletion was not controlled.
4. Pulse-chase experiments have n=1, the n should be increased and the proper statistics provided.
5. All experiments were done with overexpressed proteins that could affect the degradation rate. Currently, there are several KRAS-specific antibodies are commercially available. The effect of LZTR1 on RAS stability should be assessed on endogenous KRAS in wild-type and mutant cell lines.
6. Why do the authors conclude that RAS controls LZTR1 expression at the protein level? According to the TCGA LUAD CPTAC data, mutant KRAS is associated with increased expression of LZTR1 at mRNA but not at the protein level. The effect of KRAS mutations on mRNA levels of LZTR1 should be also checked.
7. Fig. 4E - there is no LZTR1 protein level analysis is present in this graph as claimed in the text "LZTR1 protein levels were highly upregulated in the presence of oncogenic KRAS, suggesting a possible interdependence between KRAS and LZTR1 protein levels and turnover (Fig. 4E)."
8. Fig S5B - It is confusing as CUL3 protein levels are higher in siCUL3 cells.
9. Figure legends do not always provide sufficient explanations to understand the experimental setup.

Reviewer #2 (Comments to the Authors (Required)):

The authors use APEX2 proximity labelling to identify and quantify proteins in the vicinity of KRAS and 3 different KRAS mutants. They identify both known as well as unknown potential KRAS interactors. While interactor differences between different KRAS mutants are modest, they find less LZTR1 in the KRAS mutant proximitome than in the wildtype KRAS proximitome. They also show that LZTR1 and KRAS are linked by a regulatory circle through which enhanced KRAS expression enhances LZTR1 protein levels which leads to increased KRAS degradation. Mutant KRAS proteins are less affected by LZTR1 than wildtype KRAS. T

The proteomics part of the paper is done very competently, but the functional analysis of the results is rather superficial, mainly relying on siRNA knockdown experiments. The paper would benefit from working out the new observations, e.g. the mutual regulation of KRAS and LZTR1 abundance, from a more mechanistic angle. For instance, how does KRAS increase LZTR1 abundance? Or why LZTR1 mainly affects the abundance of wildtype KRAS and not mutant KRAS (the lesser binding hypothesis is not entirely convincing, see comments below. Specific points are listed below.

Major points

Fig. 1B. It is also important to know how much the APEX tagged RAS proteins are overexpressed relative to endogenous RAS. This should be shown.

Fig. S1D. The choice of controls for the subcellular fractionation is strange. TOMM20 is a mitochondrial protein and Calreticulin mainly resides in the endoplasmic reticulum. These proteins are not really suitable to show that RAS is located at the plasma membrane as seen by microscopy.

Fig. S1E. The sucrose gradient fractionation is not very convincing. While it is true the amount of RAS shifts between fractions in response to FCS, RAS is found in all fractions of the gradient. I do not think that this is informative and that conclusions like "...APEX-2 labelling took place within the correct subcellular environment..."

Fig. 4B. "LZTR1 was detected in all samples except the beads control and was enriched in WT as compared to the mutants." LZTR1 is also less abundant in the G13D and Q61H mutant cell lines, which may well explain the lesser biotin labelling of LZTR1 in the G13D and Q61H mutant cell lines. These blots need to be quantified, and biotin labelled LZTR1 needs to be normalized to the total abundance of LZTR1 in cell extracts. It is important to clarify this point as it directly impacts on the authors' hypothesis that diminished LZTR1 binding is responsible for the greater stability of the KRAS mutant proteins.

Fig. 4C. The LZTR1 knockdown efficiency should be shown.

Fig. S5B. The downregulation of Cullin is more extensive in the control cells than in the siRNA transfected cells. While LZTR1 levels are lower in the si-Cul cells than in the si-Ctrl cells, there is no change over time in neither condition. Also, what does X mean? Is this cycloheximide treatment? This should be included in the figure legend.

Minor points

Fig. 1A. The inset in the rectangle with the broken lines is unclear. I guess it should show the transfer of biotin to the proteins in close RAS vicinity.

Lines 238-240. "Notably, the stability of KRAS is strongly diminished in the G12D and G13D KRAS mutants and slightly decreased in the Q61H KRAS mutant (Fig.4C, S5A). This suggests that the half-life of oncogenic KRAS mutants might be increased as compared to the WT counterpart." These two sentences contradict each other.

'Referee Cross-Comments' I found the other reviewers' comments fair and accurate.

Reviewer #3 (Comments to the Authors (Required)):

The authors have developed a robust system able to specifically biotinylate the KRAS proximal proteome. The ubiquitin machinery component LZTR1 is shown to be more efficiently associated with wild type KRAS than with the other mutant forms of KRAS tested. This has important implications for how RAS activity levels are boosted by activating mutations.

Overall, the paper is interesting and the data persuasive. A couple of experiments that might further support the conclusions would be:

- Overexpressing Flag-KRAS WT and "HA/other"Tag-KRAS mutant with Myc-Tag LZTR1 in the same cells and check for the half-life of both KRAS. This could nicely show LZTR1 preference for KRAS WT over mutants and strengthen the data.

- Compare LZTR1 +/- vs +/- MEFs for WT KRAS degradation (NRAS and HRAS as well...) to show it is specific to KRAS. RIT1 as a positive control (PMID: 35467524 Figure 3g - Pau Castel used RAS10 Ab). Possible to overexpress their FLAG-KRAS wt and mutants and check for half-life. If the LZTR1 MEFs are not possible, then could carry out CRISPR KO of LZTR1 in HEK293T.

Specific points:

Figure 1: 1B. Replace "IP" by GST-RAF1 RBD pulldown.

Figure 1D could be improved as it is not very easy to read, mainly because of the cut and paste WB bands which are quite heavily sloping. Making it hard to line up with markers.

In Sup Fig 1, TOMM20 is a mitochondrial membrane marker, the authors could use another membrane localisation marker as well that is more relevant for RAS (RTK, ion channel...).

Figure 2: In this figure, the authors analysed and compared their results using known databases of PPI.

I couldn't find Table S1 in the manuscript.

Figure 2C - the quality of the image is very bad and hard to read.

I think S2A and S2B should go in main figure as they are a clearer way of displaying these data.

Figure 3: Figure 3C, 3D and 3E: "(C-D) Heatmap showing the log₂(FC MUT/WT) of differentially enriched proximal proteins which were reported in previous "proximome" studies (C) or proposed based on our study (D)" Could the author highlight interactors that were found in both experiments to confirm reproducibility of the results?

Figure 4: Even if the authors previously showed that KRAS overexpression is similar upon Tet treatment (Figure 1D), they should show the expression of the KRAS wt and mutants to strengthen the point that WT KRAS associates preferably with

LZTR1, then Q61H mutant, assuming similar expression levels of the KRAS fusion proteins. Did the author blot for ARAF or RAF1 in this experiment? ARAF mentioned in the text (line 228), but can't see in the figure.

Figure 4C is a bit confusing. Could the authors show proof that LZTR1 has indeed been knocked down?

S5A is not very clear. KRAS signal is increased upon si-LZTR1, as expected. However, it also seems that there are more cells (DAPI staining) in the si-LZTR1 condition vs si-ctrl. Could the author quantify the intensity of the KRAS signal in between the 2 conditions and normalise it to DAPI or number of cells?

I think it would be important to have error bars on 4E 4F, 4G, instead of showing a representative experiment.

The results (lines 247-249) described are not shown in the figure.

Fig S5B: I think the CUL3 and Myc-Tag LZTR1 bands have been swapped (si-CUL3 condition should have reduced levels of CUL3...)

If the authors correct this issue, the figure makes sense and links CUL3 to LZTR1 as a RAS degradation complex.

POINT-BY-POINT REPLY

Reviewer #1 (Comments to the Authors (Required))

“Damianou et al. utilized the APEX-based approach to identify the proximitome of wild-type KRAS and several KRAS mutants. Even though multiple TAP-based and BioID studies focused on the characterization of RAS interactome have been published, the APEX approach allows rapid proximity labeling and could reveal dynamic associations of active and inactive forms of KRAS in response to serum stimulation. Nonetheless, the observed differences between starved and serum stimulation conditions are limited. The ubiquitin ligase LZTR1 is the only protein that showed a significant difference in the interaction between wt and mutant KRAS. In line with the interaction data, the authors showed that LZTR1 affects the stability of wt but mutant KRAS. They also proposed a feedback regulation between wt-KRAS and LZTR1 protein levels. Even though these results suggest a novel mechanism of mutant KRAS activation due to disrupted LZTR1-mediated degradation.....”

Answer: We thank the reviewer for highlighting the major findings and relevance of our work.

“...the manuscript has some major limitations that should be addressed by the authors:”

Answer: Below, we provide detailed responses to all criticisms that this reviewer raised.

“1. Fig. 4B - is problematic. pERK levels are lower upon serum stimulation of the control samples. There is no control for RAS-Apex overexpression levels that could affect the interpretation of the results.”

Answer: We acknowledge this criticism raised by the reviewer. However, we would like to emphasize that we have carefully evaluated the properties of ectopically expressed KRAS-APEX2 and demonstrate that the localisation and function are comparable to its wildtype counterpart (Figure S1). Also, to evaluate the effects on GTP hydrolysis, we examined ERK/phospho-(p)-ERK activation in all four cell lines (expressing KRAS WT, G12D, G13D and Q61H) after starvation for 2 and 4 hours, respectively. As expected, pERK levels were reduced upon starvation in WT KRAS-expressing cells. However, this reduction was not observed in the cells expressing mutant variants (Fig. 1B). Additionally, In the revised manuscript, we included a western blot validating the levels of RAS-APEX (Fig S1A) (Lines 118-119). The over-expression is between 6-8 times more when compared to endogenous KRAS levels.

A

Figure S1: Endogenous, APEX2-KRAS and GFP-KRAS association with membrane components

(A) Western blotting reveals the presence of both endogenous KRAS and exogenous KRAS APEX2 fusion proteins (WT, G12D, G13D and Q61H) in the presence of 1 μ g/mL tetracycline. The relative expression of the APEX2 KRAS fusion protein over the endogenous KRAS was determined by quantifying the bands and normalizing the data.

“2. Fig. 4A,B - Why did serum stimulation leading to the activation of KRAS increase the interaction with LZTR1, whereas activating RAS mutations inhibit the binding?”

Answer: We agree with the reviewer that this is a major point that deserves further elaboration. Our mass spectrometry and biochemical data demonstrate a greater association between activated WT KRAS with LZTR1 and less so with KRAS mutants. Although both, activated KRAS and mutants predominantly bind to GTP, this alone may not explain LZTR1 interactions per se. Molecular details of these interactions may await structural investigations, which we feel are outline the scope of this manuscript.

“3. Fig. 4C -The setup of this experiment does not make sense as the observed difference could be explained by the difference in the expression levels but not degradation. The experiment should be done like it is described in PMID: 18988848, which is considered a standard approach in the field. LZTR1 depletion was not controlled.”

Answer: We thank the reviewer for highlighting this criticism. Indeed, differences in the abundance of KRAS variants could be explained either by altered expression levels and/or degradation. It is important to point out that there are two studies that previously identified LZTR1 as a RAS regulator of its ubiquitination and signalling. Additionally, there is an additional study that has already proven the ability of LZTR1 to regulate KRAS protein stability (Abe *et al.*, 2020). This experiment was not designed to prove that LZTR1 is regulating KRAS protein stability. Our experiment was

designed to investigate if there are any differences between the WT and the mutants. Furthermore, we have performed additional experiments to determine i) KRAS & variant turnover-protein stability and ii) dependence of LZTR1. In the revised manuscript, Figure 4E-F demonstrates, using cycloheximide (CHX) chase experiments, the differential degradation rate of KRAS WT versus mutants. In the revised manuscript, we performed a Proteomic analysis where we observed that the oncogenic mutants (G12D and G13D) of KRAS have higher protein levels compared to the WT (Fig. 4B, Fig S3F-G) (lines 218-238). For the second point, Figure 4G, S5C demonstrate the greater susceptibility of wildtype KRAS to LZTR1 levels. Additionally, a CHX experiment in the presence or absence of LZTR1 was performed, indicating that WT half-life is markedly affected by the presence of LZTR1 as compared to the mutants (Fig. 4H). We included in the revised manuscript Figure S5B where we evaluated LZTR1 knockdown efficiency during the performance of the microscopy experiment (Fig 4C). To improve clarity, we amended the results section accordingly (lines 241-247).

Fig. 4A-Fig. S3F-G: Proteomic Analysis of G12D, G13D, and Q61H compared to KRAS WT

Volcano plotting of fold changes and p values derived from t-test statistic total proteome identified in G12D, G13D, and Q61H KRAS mutant in starvation, in which WT-KRAS was used as a control. Known KRAS effectors (red), repressors (green), receptors (blue) and KRAS (black) are coloured and labelled.

“4. Pulse-chase experiments have n=1, the n should be increased and the proper statistics provided.”

Answer: The reviewer refers to Figure 4F and H concerning this comment. To clarify, what is shown here are CHX chase experiments (not pulse-chase), including multiple time point. N=1 is acceptable if there are many time points taken that show considerable differences between wildtype KRAS and mutants in terms of abundance.

“5. All experiments were done with overexpressed proteins that could affect the degradation rate. Currently, there are several KRAS-specific antibodies are commercially available. The

effect of LZTR1 on RAS stability should be assessed on endogenous KRAS in wild-type and mutant cell lines.”

Answer: We appreciate this comment made by the reviewer. In an ideal case scenario, cell lines expressing either wild-type KRAS or mutant forms (G12D, G13D, Q61H) at the endogenous level should be explored in this context. However, we do not have access to such cell lines, in particular, those with two mutant alleles or whether the wild-type counterpart has been deleted. To address this point, at least in part, we have measured ectopic expression levels and found them to be 6- to 8-fold above endogenous levels (revised Figure S1A). Also, we demonstrate that ectopically expressed KRAS does behave like its endogenous counterpart regarding localisation and function (Fig 1B and S1E) (see also response to section 1.).

“6. Why do the authors conclude that RAS controls LZTR1 expression at the protein level? According to the TCGA LUAD CPTAC data, mutant KRAS is associated with increased expression of LZTR1 at mRNA but not at the protein level. The effect of KRAS mutations on mRNA levels of LZTR1 should be also checked.”

Answer: We thank the reviewer for raising this important detail. We have complied with this and explored the effect of KRAS variants on LZTR1 mRNA levels (revised Figure 5G). We found that LZTR1 mRNA levels remained unaltered upon induction of either WT KRAS or the three mutant variants, thereby suggesting that any observed changes are not a consequence of mRNA elevation (lines 301-307).

Figure S5: Interdependence of LZTR1 and KRAS protein stabilities controlled by CUL3

(G)Relative mRNA expression level of LZTR1 in different cell lines (WT, G12D, G13D, and Q61H) without or with treatment of 1 µg/mL tetracycline overnight. Relative values are normalized by β-actin. Error bars represent S.E.M of at least three independent experiments.

“7. Fig. 4E - there is no LZTR1 protein level analysis is present in this graph as claimed in the text “LZTR1 protein levels were highly upregulated in the presence of oncogenic KRAS, suggesting a possible interdependence between KRAS and LZTR1 protein levels and turnover (Fig. 4E).”

We apologize for the oversight. We have now corrected this in the revised version (Figure 4G). In other experiments, we also observed that LZTR1 protein levels were higher in WT KRAS compared to mutated KRAS. For instance, in Figure I and J, in the absence of MLN4924 or at time 0, this difference was observed.

Furthermore, in the revised manuscript, we conducted a proteomic analysis of the KRAS APEX2 WT, G12D, G13D, and Q61H cell lines, monitoring the levels of LZTR1 (lines 218-238). We found that LZTR1 endogenous protein levels were significantly upregulated, in the case of the Q61H mutation compare to WT (Fig S3G). To ensure that this effect was not due to increased LZTR1 mRNA levels, we performed RT-PCR, which showed no upregulation of LZTR1 mRNA levels (Fig S5G) (lines 301-307).

The Proteomic analysis indicated that the endogenous levels of LZTR1 are not significantly altered in the presence of G12D and G13D mutants. We believe that this may be due to the establishment of a cross-talk equilibrium between LZTR1 and KRAS in that scenario. Additionally, it's important to note that all experiments monitoring the effect of KRAS on LZTR1 protein levels were conducted as ectopic experiments.

We have revised our discussion section to elaborate on the impact of KRAS on LZTR1 in both scenarios. It's crucial to clarify that the primary focus of our study lies in the differences observed in the protein degradation of oncogenic KRAS, which is influenced by LZTR1. Regarding the cross-talk between KRAS and LZTR1, we have described our observations and provided a possible explanation. We acknowledge that further exploration of this complex interplay is warranted and should be undertaken by experts in KRAS biology equipped with the appropriate tools (lines 363-377).

“8. Fig S5B - It is confusing as CUL3 protein levels are higher in siCUL3 cells.”

Answer: We apologise for this error. This has now been corrected in the revised version (revised manuscript, Figure S5F).

“9. Figure legends do not always provide sufficient explanations to understand the experimental setup.”

Answer: We acknowledge this criticism made by the reviewer. To improve clarity, we have now included additional explanations in the Figure legend section, especially for Figures 2, 4, S1, S3 and S5; (pages 25, 26, 28, 29 and 30).

Reviewer #2 (Comments to the Authors (Required)):

“The authors use APEX2 proximity labelling to identify and quantify proteins in the vicinity of KRAS and 3 different KRAS mutants. They identify both known as well as unknown potential KRAS interactors. While interactor differences between different KRAS mutants are modest, they find less LZTR1 in the KRAS mutant proximitome than in the wildtype KRAS proximitome. They also show that LZTR1 and KRAS are linked by a regulatory circle through which enhanced KRAS expression enhances LZTR1 protein levels which leads to increased KRAS degradation. Mutant KRAS proteins are less affected by LZTR1 than wildtype KRAS.

*T
The proteomics part of the paper is done very competently...”*

Answer: We thank the reviewer for the generally positive evaluation of our work.

“... but the functional analysis of the results is rather superficial, mainly relying on siRNA knockdown experiments. The paper would benefit from working out the new observations, e.g. the mutual regulation of KRAS and LZTR1 abundance, from a more mechanistic angle. For instance, how does KRAS increase LZTR1 abundance? Or why LZTR1 mainly affects the abundance of wildtype KRAS and not mutant KRAS (the lesser binding hypothesis is not entirely convincing, see comments below. Specific points are listed below.”

Answer: We appreciate that the reviewer seems excited about our original observations regarding a functional interplay between KRAS and LZTR1 that seems altered by KRAS mutations. Our manuscript aimed to provide an initial biochemical characterisation of this interrelationship that was originally proposed by the proximity labelling experiment. However, the exact nature of how this works on a detailed molecular level might warrant future investigations. At this point, we feel that this is currently outside the scope of our manuscript.

“Major points

Fig. 1B. It is also important to know how much the APEX tagged RAS proteins are overexpressed relative to endogenous RAS. This should be shown.”

Answer: We agree that this is a valid concern. To comply, we addressed this in our experiment (revised Figure S1A), where we showed that the APEX2-tagged KRAS exhibits an overexpression ranging from 4 to 8-fold compared to the endogenous RAS. See also amended text in our revised manuscript (lines 681-697).

A

Figure S1A: Endogenous, APEX2-KRAS and GFP-KRAS association with membrane components

Western blotting reveals the presence of both endogenous KRAS and exogenous KRAS APEX2 fusion proteins (WT, G12D, G13D and Q61H) in the presence of 1 μ g/mL tetracycline. The relative expression of the APEX2 KRAS fusion protein over the endogenous KRAS was determined by quantifying the bands and normalizing the data.

“Fig. S1D. The choice of controls for the subcellular fractionation is strange. TOMM20 is a mitochondrial protein and Calreticulin mainly resides in the endoplasmic reticulum. These proteins are not really suitable to show that RAS is located at the plasma membrane as seen by microscopy.”

Answer: We agree with the reviewer that TOMM20 may not be suitable to show the KRAS localization. Thus, in the revised manuscript, we replaced the TOMM20 western blot with a Na,K-ATPase blot (Figure S1D, lines 692-695), a widely recognized plasma membrane marker, which also aligns with our microscopic result.

D

Figure S1D: Endogenous, APEX2-KRAS and GFP-KRAS association with membrane components

Western blot analysis showing a-FLAG, endogenous KRAS, and Na, K-ATPase expression upon digitonin cytoplasmic-membrane fractionation of stable KRAS APEX2 WT, G12D, G13D and Q61H FRT T-Rex HEK293 cell lines. Cells were incubated with or without 1 µg/mL tetracycline for 24 hours.

"Fig. S1E. The sucrose gradient fractionation is not very convincing. While it is true the amount of RAS shifts between fractions in response to FCS, RAS is found in all fractions of the gradient. I do not think that this is informative and that conclusions like "...APEX-2 labelling took place within the correct subcellular environment..."

Answer: We acknowledge this criticism raised by the reviewer. In response, we would like to clarify that while both endogenous KRAS and APEX-KRAS can be detected in low levels across all fractions, our intention was to illustrate a shift in their distribution, particularly towards heavier fractions upon FCS induction. The main purpose of this experiment was to demonstrate the translocation of both endogenous and exogenously overexpressed RAS. To improve clarity in the revised version of the manuscript, we have updated FigS1E and adjusted the related sections (lines 138-142) in the main text to better convey this point.

"Fig. 4B. "LZTR1 was detected in all samples except the beads control and was enriched in WT as compared to the mutants." LZTR1 is also less abundant in the G13D and Q61H mutant cell lines, which may well explain the lesser biotin labelling of LZTR1 in the G13D and Q61H mutant cell lines. These blots need to be quantified, and biotin labelled LZTR1 needs to be normalized to the total abundance of LZTR1 in cell extracts. It is important to clarify this point as it directly impacts on the authors' hypothesis that diminished LZTR1 binding is responsible for the greater stability of the KRAS mutant proteins."

Answer: We thank the reviewer for raising this criticism. Since we found it challenging to find a high-quality antibody for LZTR1, we performed a deep proteomic analysis to quantify LZTR1's protein levels. Our results showed that LZTR1 protein levels remained the same between the wild-type (WT) and G12D and G13D. On the other hand, We could monitor that LZTR1 protein levels were significantly enriched in the Q61H mutants compared to the WT. Nevertheless, in the proximity labelling experiment, the LZTR1 was found to be enriched in the WT compared to the Q61H mutation. In light of this finding, we replaced Figure 4B with a scatter plot that integrates the proteomic and proximity labelling data. After normalization by proteomic data, this new panel clearly demonstrates that differentially enriched LZTR1 in the proximity proteome of WT and mutant KRAS is not caused by LZTR1 abundance change at the protein level. To reflect this, we have revised the text in the results section (lines 218-244).

Integration of the “proximitome” and proteome.

(H-I) Scatter plot showing the integration of the "proximitome" and proteome of G12D, G13D and Q61H. On the Y-axis, the fold change between the mutant and the WT for the "proximitome" is represented. On the X-axis, the fold change between the mutant and the WT in the proteome experiment is illustrated. Known KRAS effectors are highlighted in red, repressors in green, receptors in blue, and KRAS in black with corresponding labels.

“Fig. 4C. The LZTR1 knockdown efficiency should be shown.”

Answer: Thank you for your suggestion. To address this concern, we have included a WB result (FigS5B) showing LZTR1 knockdown efficiency using the same samples employed in the microscopic experiment.

Figure S5: Interdependence of LZTR1 and KRAS protein stabilities controlled by CUL3

Following the microscopy conducted in Fig 4D and S5A, cells were lysed directly in the 96-well plate, and a Western blot analysis was subsequently performed to assess the effectiveness of LZTR1 knockdown.

"Fig. S5B. The downregulation of Cullin is more extensive in the control cells than in the siRNA transfected cells. While LZTR1 levels are lower in the si-Cul cells than in the si-Ctrl cells, there is no change over time in neither condition. Also, what does X mean? Is this cycloheximide treatment? This should be included in the figure legend."

Answer: We thank the reviewer for flagging up these technical points. LZTR1 levels appear to also depend on CUL3 protein expression, further stressing a possible relationship between CUL3-LZTR1 and KRAS.

Regarding to the "X" label, we apologise for this oversight as this should mean CHX (Cycloheximide). To improve clarification, we have addressed these concerns in revised Fig. S5D, reflecting the corrected information regarding this figure.

"Minor points

Fig. 1A. The inset in the rectangle with the broken lines is unclear. I guess it should show the transfer of biotin to the proteins in close RAS vicinity.

Answer: Thank you for your comment. In the revised version, we removed the rectangle with the broken lines so we don't confuse the readers.

"Lines 238-240. "Notably, the stability of KRAS is strongly diminished in the G12D and G13D KRAS mutants and slightly decreased in the Q61H KRAS mutant (Fig.4C, S5A). This suggests that the half-life of oncogenic KRAS mutants might be increased as compared to the WT counterpart." These two sentences contradict each other."

Answer: We thank the reviewer for pointing out this discrepancy. We have now rephrased this statement to improve clarity (lines 248-249).

'Referee Cross-Comments' I found the other reviewers' comments fair and accurate.

Reviewer #3 (Comments to the Authors (Required)):

"The authors have developed a robust system able to specifically biotinylate the KRAS proximal proteome. The ubiquitin machinery component LZTR1 is shown to be more efficiently associated with wild type KRAS than with the other mutant forms of KRAS tested. This has important implications for how RAS activity levels are boosted by activating mutations."

"Overall, the paper is interesting and the data persuasive. A couple of experiments that might further support the conclusions would be:"

Answer: We thank the reviewer for emphasizing the key discovery and significance of our study.

“- Overexpressing Flag-KRAS WT and "HA/other"Tag-KRAS mutant with Myc-Tag LZTR1 in the same cells and check for the half-life of both KRAS. This could nicely show LZTR1 preference for KRAS WT over mutants and strengthen the data.”

Answer: Thank you, the reviewer, for your suggestion. We decided to perform that experiment. We orchestrated an over-expression of GFP-KRAS fusion proteins (WT and G12D) in the concurrent presence or absence of LZTR1 while monitoring both endogenous WT KRAS and exogenous GFP KRAS (Fig S5C). In this experiment, we could easily see the preference of LZTR1 towards endogenous WT KRAS compared to the G12D KRAS mutated version, and this was not monitored when we over-expressed WT KRAS (lines 253-260).

Figure S5: Interdependence of LZTR1 and KRAS protein stabilities controlled by CUL3

FLP-in stable HEK293 cell lines expressing either GFP WT or G12D were seeded in 6-well plates and subjected to 1 µg/mL tetracycline treatment for 24 hours. Subsequently, the following day, the cells were transfected with LZTR1 siRNA or Control siRNA. The cells were cultured for an additional 48 hours before assessing protein levels through Western blot analysis.

“- Compare LZTR1 +/+ vs -/- MEFs for WT KRAS degradation (NRAS and HRAS as well...) to show it is specific to KRAS. RIT1 as a positive control (PMID: 35467524 Figure 3g - Pau Castel used RAS10 Ab). Possible to overexpress their FLAG-KRAS wt and mutants and check for half-life. If the LZTR1 MEFs are not possible, then could carry out CRISPR KO of LZTR1 in HEK293T.”

Answer: We appreciate this comment from the reviewer. It will be great to evaluate those results in the KO or LZTR1 MEF. However, due to the limitation of time, this could be something that people in the KRAS area could follow up on.

“Specific points:”

“Figure 1: 1B. Replace "IP" by GST-RAF1 RBD pulldown.”

Answer: Thank you for your comment. Please check the revised figure where we replaced IP with GST-RAF1 RBD.

“Figure 1D could be improved as it is not very easy to read, mainly because of the cut and paste WB bands which are quite heavily sloping. Making it hard to line up with markers. In Sup Fig 1, TOMM20 is a mitochondrial membrane marker, the authors could use another membrane localisation marker as well that is more relevant for RAS (RTK, ion channel...)”

Answer: We appreciate the reviewer’s attention to these specific points. We have accordingly made amendments for Fig 1B and 1D. Furthermore, in line with our response to the reviewer, we agree that TOMM20 may not be suitable to show the KRAS localization. Thus, in the revised manuscript, we replaced the TOMM20 western blot with a Na,K-ATPase blot (Figure S1D, lines 692-695), a widely recognized plasma membrane marker, which also aligns with our microscopic result.

D

Figure S1D: Endogenous, APEX2-KRAS and GFP-KRAS association with membrane components

Western blot analysis showing a-FLAG, endogenous KRAS, and Na, K-ATPase expression upon digitonin cytoplasmic-membrane fractionation of stable KRAS APEX2 WT, G12D, G13D and Q61H FRT T-Rex HEK293 cell lines. Cells were incubated with or without 1 µg/mL tetracycline for 24 hours.

“Figure 2: In this figure, the authors analysed and compared their results using known databases of PPI.

I couldn't find Table S1 in the manuscript.

Figure 2C - the quality of the image is very bad and hard to read.

I think S2A and S2B should go in main figure as they are a clearer way of displaying these data.”

Answer: We apologize for the oversight and have now added Table S1 to the manuscript. Additionally, we have removed Fig 2C as we believe it does not provide significant additional information to the revised manuscript.

Regarding Fig S2A and S2B, we have included them as supplementary figures because the bead control is less than ideal due to the high background of APEX2. By

comparing the different mutants with the WT, we were able to mitigate most of the background associated with the proximity labelling approach.

“Figure 3: Figure 3C, 3D and 3E: “(C-D) Heatmap showing the $\log_2(\text{FC MUT/WT})$ of differentially enriched proximal proteins which were reported in previous “proximitome” studies (C) or proposed based on our study (D)” Could the author highlight interactors that were found in both experiments to confirm reproducibility of the results?”

Answer: Thank you for your suggestion. We include a heat map showing the $\log_2(\text{FC MUT/WT})$ for the Classical RAS effectors and Repressors (Fig S2D and E, lines 188-193). In this figure, we could show nicely the significant enriched of RAF1 and ARAF in the mutations compared to the WT.

Figure S2: Enriched proteomes between KRAS wildtype exposed to starvation or acute FCS stimulation
 (D) Heatmap showing the $\log_2(\text{FC MUT/WT})$ of Classical RAS Effectors which are differentially enriched proximal proteins in our study. An asterisk (*) indicates that the hit meets the criteria of $\log_2(\text{mutant/WT FC}) > 0.5$ or < -0.5 and $-\log_{10}(\text{p value}) > 0.7$.

“Figure 4: Even if the authors previously showed that KRAS overexpression is similar upon Tet treatment (Figure 1D), they should show the expression of the KRAS wt and mutants to strengthen the point that WT KRAS associates preferably with LZTR1, then Q61H mutant, assuming similar expression levels of the KRAS fusion proteins. Did the author blot for ARAF or RAF1 in this experiment? ARAF mentioned in the text (line 228), but can't see in the figure.”

Answer: To address these concerns, we conducted a mass spectrometry (MS) proteomic analysis, enabling us to monitor both KRAS and LZTR1 levels. The volcano plot comparing WT and the different mutants confirmed our expectations, showing that KRAS mutations are associated with increased protein stability compared to the WT.

To account for variations in labelling across different conditions, we normalized the proximitome samples to the total Streptavidin signal following Streptavidin immunoprecipitation (IP). In our proteomic experiment, we observed that the levels of other KRAS-interacting partners remained relatively stable between WT and mutants, with LZTR1 showing minimal changes in most cases. An exception was found when comparing WT to the Q61H mutant, where higher LZTR1 levels were observed in the Q61H mutant compared to the WT.

Interestingly, the results from the “proximitome” analysis showed a reversal, indicating higher LZTR1 levels in the WT compared to the Q61H mutant. This discrepancy suggests that the differences between the WT and mutant “proximitomes” might be even more pronounced if LZTR1 levels were comparable between them.

We acknowledge the reviewer's concern regarding APEX2 fusion protein levels. Normalizing the Streptavidin signal was a deliberate strategy to account for fluctuations in proximity labeling and variations in the expression of different RAS-APEX2 variants. Notably, our observations, particularly in the case of LZTR1, a key hit validating our study, revealed higher levels of KRAS in the Q61H mutant and more LZTR1 in the proximitome of the WT. This intriguing reversal in the results adds confidence to our conclusion that the changes in LZTR1 abundance within the proximitome are not solely attributed to higher expression of KRAS-APEX2 (lines 218-238).

Fig. 4A-Fig. S3F-G: Proteomic Analysis of G12D, G13D, and Q61H compared to KRAS WT

Volcano plotting of fold changes and p values derived from t-test statistic total proteome identified in G12D, G13D, and Q61H KRAS mutant in starvation, in which WT-KRAS was

used as a control. Known KRAS effectors (red), repressors (green), receptors (blue) and KRAS (black) are coloured and labelled.

Integration of the "proximitome" and proteome.

(H-I) Scatter plot showing the integration of the "proximitome" and proteome of G12D, G13D and Q61H. On the Y-axis, the fold change between the mutant and the WT for the "proximitome" is represented. On the X-axis, the fold change between the mutant and the WT in the proteome experiment is illustrated. Known KRAS effectors are highlighted in red, repressors in green, receptors in blue, and KRAS in black with corresponding labels.

"Figure 4C is a bit confusing. Could the authors show proof that LZTR1 has indeed been knocked down?"

Answer: Thank you. This has now been clarified in revised Fig S5B.

"S5A is not very clear. KRAS signal is increased upon si-LZTR1, as expected. However, it also seems that there are more cells (DAPI staining) in the si-LZTR1 condition vs si-ctrl. Could the author quantify the intensity of the KRAS signal in between the 2 conditions and normalise it to DAPI or number of cells?"

I think it would be important to have error bars on 4E 4F, 4G, instead of showing a representative experiment."

Answer: Your concern is entirely valid, and we sincerely appreciate your feedback. We acknowledge that it may not have been apparent that those images were intended as examples used in the generation of Fig 4C, where the signal was normalized based on DAPI staining. To address this, we revised the figure legend to provide clearer and more explicit information in the new version of our work.

"The results (lines 247-249) described are not shown in the figure.

Fig S5B: I think the CUL3 and Myc-Tag LZTR1 bands have been swapped (si-CUL3 condition should have reduced levels of CUL3...)

If the authors correct this issue, the figure makes sense and links CUL3 to LZTR1 as a RAS degradation complex."

Answer: We have addressed the concern, and the issue with Fig. S5B has been resolved. We have now also updated Fig. S5D and included the corrected information regarding this figure.

January 25, 2024

RE: Life Science Alliance Manuscript #LSA-2023-02245-TR

Dr. Andreas Damianou

University of Oxford

Target Discovery Institute, Centre for Medicines Discovery, Nuffield Department of Medicine, University of Oxford, Roosevelt Drive, Oxford OX3 7FZ, UK
United Kingdom

Dear Dr. Damianou,

Thank you for submitting your revised manuscript entitled "Oncogenic Mutations of KRAS Modulate Its Turnover by the CUL3/LZTR1 E3 ligase Complex". We would be happy to publish your paper in Life Science Alliance pending final revisions necessary to meet our formatting guidelines.

- please address Reviewer 1's remaining comments
- please be sure that the authorship listing and order is correct
- please add a Category for your manuscript in our system
- please be sure that all authors are included in the Author Contribution section in the manuscript file
- please add your main, supplementary figure, and table legends to the main manuscript text after the references section
- please add callouts for Figure S4A-D to your main manuscript text

A. FINAL FILES:

B. MANUSCRIPT ORGANIZATION AND FORMATTING:

**Submission of a paper that does not conform to Life Science Alliance guidelines will delay the acceptance of your

manuscript.**

The license to publish form must be signed before your manuscript can be sent to production. A link to the electronic license to publish form will be available to the corresponding author only. Please take a moment to check your funder requirements.

Sincerely,

Reviewer #1 (Comments to the Authors (Required)):

The authors have provided additional controls and explanations that improved the manuscript.

- validated that APEX2 tagging did not impact the function of the mutant and wild-type KRAS-APEX2 fusion protein, using immunoblotting for Phospho-ERK1/2, RAF1-RBD pulldown, and KRAS translocation assessment.
- clarified the list of KRAS interactors detected, compared to the one previously detected, showing that 21% were detected here.
- clarified the impact of KRAS mutant- APEX2 on the interactome with proteins associated to the Phospho-ERK1/2 pathway,
- demonstrated that LZTR1 showed a weaker association with oncogenic G12D, G13D, and Q61H KRAS mutants when compared to WT, associated with a reduction of KRAS-KRAS interaction
- performed proteome analysis of cells overexpressing either WT or different KRAS mutants.

However, several points still need to be further clarified:

1. Neither main nor supplemental figures are labeled, it is difficult to find the right figure.
2. Fig S3 is difficult to follow. A and F as well as B and G look the same.
3. New Fig 1B - why did WT-RAS overexpression reduce the levels of ERK1/2 phosphorylation? Phospho-ERK1/2 phosphorylation was not assessed upon serum stimulation.
4. I would disagree that N=1 is acceptable if there are many time points taken. It would mean that N=1 will be enough to show, for example, tumor growth or therapeutic response over time.
5. Only 20% of known KRAS were detected here, this seems relatively low. Could this be discussed? Can the authors further compare their results with the ones observed with previous KRAS, TRAP, Bio-ID, or Turbo-ID interactome studies? Could this low detection rate be due to the limitations of the APEX approach?
6. The observation that LZTR1 showed a weaker association with oncogenic G12D, G13D, and Q61H KRAS mutants when compared to active WT should be further discussed.
7. Based on the new proteomic data, the claim regarding the interplay between LZTR1 and RAS levels sounds like an overstatement.

Reviewer #2 (Comments to the Authors (Required)):

The author have addressed my queries satisfactorily, and the revised paper is suitable for publication.

Reviewer #3 (Comments to the Authors (Required)):

I am happy with the revisions made to this paper and believe that it is now suitable for publication in Life Science Alliance.

POINT-BY-POINT REPLY R2**Reviewer #1 (Comments to the Authors (Required)):**

The authors have provided additional controls and explanations that improved the manuscript.

- validated that APEX2 tagging did not impact the function of the mutant and wild-type KRAS-APEX2 fusion protein, using immunoblotting for Phospho-ERK1/2, RAF1-RBD pulldown, and KRAS translocation assessment.
- clarified the list of KRAS interactors detected, compared to the one previously detected, showing that 21% were detected here.
- clarified the impact of KRAS mutant- APEX2 on the interactome with proteins associated to the Phospho-ERK1/2 pathway,
- demonstrated that LZTR1 showed a weaker association with oncogenic G12D, G13D, and Q61H KRAS mutants when compared to WT, associated with a reduction of KRAS-KRAS interaction
- performed proteome analysis of cells overexpressing either WT or different KRAS mutants.

We thank the reviewer for acknowledging the amendments we made to our manuscript's revised version.

However, several points still need to be further clarified:

1. Neither main nor supplemental figures are labelled; it is difficult to find the right figure.

We thank the reviewer for raising this point. We apologise for this oversight. To improve clarity, we have added appropriate figure labels.

2. Fig S3 is difficult to follow. A and F as well as B and G look the same.

We thank the reviewer for spotting this. To improve clarity, we have added appropriate figure labels.

3. New Fig 1B - why did WT-RAS overexpression reduce the levels of ERK1/2 phosphorylation? Phospho-ERK1/2 phosphorylation was not assessed upon serum stimulation.

We acknowledge this reviewer's criticism of this experimental detail. The reason is that we collected cells at a steady state after a period of starvation to reduce phosphorylation levels, and there might be fluctuations in ERK phosphorylation at baseline. We assessed Phospho-ERK1/2 phosphorylation upon serum stimulation in Fig 1D.

4. I would disagree that N=1 is acceptable if there are many time points taken. It would mean that N=1 will be enough to show, for example, tumor growth or therapeutic response over time.

We understand this reviewer's point about statistical analysis but would argue that this is an inappropriate comparison. Recognizing that one of the main findings of our study revolves around the changes in KRAS protein half-life, we conducted an additional cycloheximide (CHX) experiment with extended treatment duration, this time in triplicate. This experiment provided a more comprehensive understanding of KRAS and LZTR1 half-life dynamics. This new data has now been included in revised Fig. S5D-E.

Figure S5: Oncogenic mutations of KRAS modulate its turnover by LZTR1

HEK293T cells were transfected with 1 μ g Flag-RAS and 0.2 μ g Myc-LZTR1 plasmid concentration. Cells were incubated for 48 hours, treated with 50 μ g/mL cycloheximide (CHX) and harvested at the indicated time points. Western blot analysis of FLAG (KRAS), MYC (LZTR1), and ACTB in cell extract are shown.

5. Only 20% of known KRAS were detected here, this seems relatively low. Could this be discussed? Can the authors further compare their results with the ones observed with previous KRAS, TRAP, Bio-ID, or Turbo-ID interactome studies? Could this low detection rate be due to the limitations of the APEX approach?

The reason why only 20% of known KRAS interacting proteins were detected may be ascribed to multiple factors, not solely due to the constraints of the APEX2 approach. One potential factor could be the absence of certain proteins in the cell line HEK293. A previous Bio-ID study by Kovalski *et al.*, 2019, identified 11 RAS interacting proteins (**Fig 1D**) compared to the 22 proteins we detected. The key difference between APEX2 and other proximity methods is the labelling time (timescale of minutes for APEX2, and hours for other proximity strategies). Additionally, the presence of H₂O₂ in APEX2 could potentially affect cellular processes. In our results section, we have already compared and mentioned that we detected 206 proteins previously identified in Bio-ID studies (**Line 170**). We further explain the absence of some canonical proteins (**Line 357**). Moreover, in our discussion, we described the limitations of our approach and compared it to some previous Bio-ID studies (**Lines 339-359**). We acknowledge the reviewer's suggestion to discuss the low detection rate of known KRAS proteins and to compare our results with previous KRAS, TRAP, Bio-ID, or Turbo-ID interactome studies. However, to better characterize the differences between APEX2 and Bio-ID, we believe that it would be more appropriate to compare these two proximity strategies in the same cell line and employing the same mass spectrometry instrumentation/methodology.

6. The observation that LZTR1 showed a weaker association with oncogenic G12D, G13D, and Q61H KRAS mutants when compared to active WT should be further discussed.

We thank the reviewer for this suggestion. We have included this point in the discussion section (**lines 374-388**), which we have now expanded to emphasise this observation further.

7. Based on the new proteomic data, the claim regarding the interplay between LZTR1 and RAS levels sounds like an overstatement.

We acknowledge the reviewer's criticism. To address this issue, we removed the wording "interplay" in the manuscript and the title and toned down the conclusions. Additionally, our new CHX experiment has revealed an effect of KRAS on LZTR1 protein turnover, which is an aspect that cannot be answered solely by proteomic experiments. Specifically, we observed that the protein half-life of LZTR1 is affected in the presence of WT KRAS. However, it remains unclear whether this effect is solely due to the low expression of KRAS (WT or mutated) or is influenced by the presence of the mutation.

Reviewer #2 (Comments to the Authors (Required)):

The author have addressed my queries satisfactorily, and the revised paper is suitable for publication.

We are grateful for this reviewer to recommend our revised manuscript for publication.

Reviewer #3 (Comments to the Authors (Required)):

I am happy with the revisions made to this paper and believe that it is now suitable for publication in Life Science Alliance.

We are grateful for this reviewer to recommend our revised manuscript for publication.

Editorial Comments:

-please address Reviewer 1's remaining comments

Please see detailed Point-by-Point Reply above

-please be sure that the authorship listing and order is correct

OK, checked

-please add a Category for your manuscript in our system

OK, provided

-please be sure that all authors are included in the Author Contribution section in the manuscript file

OK, checked

-please add your main, supplementary figure, and table legends to the main manuscript text after the references section

OK, done

-please add callouts for Figure S4A-D to your main manuscript text

OK, provided

OK

LSA now encourages authors to provide a 30-60 second video where the study is briefly explained. We will use these videos on social media to promote the published paper and the presenting author (for examples, see <https://twitter.com/LSAjournal/timelines/1437405065917124608>).

Corresponding or first-authors are welcome to submit the video. Please submit only one video per manuscript. The video can be emailed to contact@life-science-alliance.org

To upload the final version of your manuscript, please log in to your account:
<https://lsa.msubmit.net/cgi-bin/main.plex>

OK, done

OK

A. FINAL FILES:

OK, provided

OK, provided

“KRAS is a proto-oncogene encoding a small GTPase, which is mutated in ~30% of human solid tumours. Oncogenic mutations affect KRAS turnover by the CUL3/LZTR1 E3 ligase complex.” (179 characters)

B. MANUSCRIPT ORGANIZATION AND FORMATTING:

OK, happy to be included and published alongside the manuscript

OK

Sincerely,

February 27, 2024

RE: Life Science Alliance Manuscript #LSA-2023-02245-TRR

Dr. Andreas Damianou
University of Oxford
Target Discovery Institute, Centre for Medicines Discovery, Nuffield Department of Medicine, University of Oxford
Roosevelt Drive, Oxford OX3 7FZ, UK
Oxford, Oxfordshire OX3 7FZ
United Kingdom

Dear Dr. Damianou,

Thank you for submitting your Research Article entitled "Oncogenic Mutations of KRAS Modulate Its Turnover by the CUL3/LZTR1 E3 ligase Complex". It is a pleasure to let you know that your manuscript is now accepted for publication in Life Science Alliance. Congratulations on this interesting work.

DISTRIBUTION OF MATERIALS:

Again, congratulations on a very nice paper. I hope you found the review process to be constructive and are pleased with how the manuscript was handled editorially. We look forward to future exciting submissions from your lab.

Sincerely,
